# DiffWire: Inductive Graph Rewiring via the Lovász Bound

**Adrian Arnaiz-Rodriguez**
ELLIS Alicante
adrian@ellisalicante.org

**Ahmed Begga**
University of Alicante

**Francisco Escolano**
ELLIS Alicante
sco@ellisalicante.org

**Nuria Oliver**
ELLIS Alicante
nuria@ellisalicante.org

## Abstract

Graph Neural Networks (GNNs) have been shown to achieve competitive results to tackle graph-related tasks, such as node and graph classification, link prediction and node and graph clustering in a variety of domains. Most GNNs use a message passing framework and hence are called MPNNs. Despite their promising results, MPNNs have been reported to suffer from over-smoothing, over-squashing and under-reaching. Graph rewiring and graph pooling have been proposed in the literature as solutions to address these limitations. However, most state-of-the-art graph rewiring methods fail to preserve the global topology of the graph, are neither differentiable nor inductive, and require the tuning of hyper-parameters. In this paper, we propose DIFFWIRE, a novel framework for graph rewiring in MPNNs that is principled, fully differentiable and parameter-free by leveraging the Lovász bound. The proposed approach provides a unified theory for graph rewiring by proposing two new, complementary layers in MPNNs: CT-LAYER, a layer that learns the commute times and uses them as a relevance function for edge re-weighting; and GAP-LAYER, a layer to optimize the spectral gap, depending on the nature of the network and the task at hand. We empirically validate the value of each of these layers separately with benchmark datasets for graph classification. We also perform preliminary studies on the use of CT-LAYER for homophilic and heterophilic node classification tasks. DIFFWIRE brings together the learnability of commute times to related definitions of curvature, opening the door to creating more expressive MPNNs.

## 1 Introduction

Graph Neural Networks (GNNs) [1, 2] are a class of deep learning models applied to graph structured data. They have been shown to achieve state-of-the-art results in many graph-related tasks, such as node and graph classification [3, 4], link prediction [5] and node and graph clustering [6, 7], and in a variety of domains, including image or molecular structure classification, recommender systems and social influence prediction [8].

Most GNNs use a message passing framework and thus are referred to as Message Passing Neural Networks (MPNNs) [4] . In these networks, every node in each layer receives a message from its adjacent neighbors. All the incoming messages at each node are then aggregated and used to update the node's representation via a learnable non-linear function –which is typically implemented by means of a neural network. The final node representations (called node embeddings) are used to perform the graph-related task at hand (e.g. graph classification). MPNNs are extensible, simple and have proven to yield competitive empirical results. Examples of MPNNs include GCN [3], GAT [9], GATv2 [10], GIN [11] and GraphSAGE [12]. However, they typically use transductive learning, i.e. the model observes both the training and testing data during the training phase, which might limit their applicability to graph classification tasks.

A. Arnaiz-Rodriguez et al., DiffWire: Inductive Graph Rewiring via the Lovász Bound. *Proceedings of the First Learning on Graphs Conference (LoG 2022)*, PMLR 198, Virtual Event, December 9–12, 2022.

However, MPNNs also have important limitations due to the inherent complexity of graphs. Despite such complexity, the literature has reported best results when MPNNs have a small number of layers, because networks with many layers tend to suffer from *over-smoothing* [13] and *over-squashing* [14]. However, this models fail to capture information that depends on the entire structure of the graph [15] and prevent the information flow to reach distant nodes. This phenomenon is called *under-reaching* [16] and occurs when the MPNN's depth is smaller than the graph's diameter.

*Over-smoothing* [8, 17–19] takes place when the embeddings of nodes that belong to different classes become indistinguishable. It tends to occur in MPNNs with many layers that are used to tackle short-range tasks, i.e. tasks where a node's correct prediction mostly depends on its local neighborhood. Given this local dependency, it makes intuitive sense that adding layers to the network would not help the network's performance.

Conversely, long-range tasks require as many layers in the network as the range of the interaction between the nodes. However, as the number of layers in the network increases, the number of nodes feeding into each of the node's receptive field also increases exponentially, leading to *over-squashing* [14, 20]: the information flowing from the receptive field composed of many nodes is compressed in fixed-length node vectors, and hence the graph fails to correctly propagate the messages coming from distant nodes. Thus, over-squashing emerges due to the distortion of information flowing from distant nodes due to graph bottlenecks that emerge when the number of $k$-hop neighbors grows exponentially with $k$.

Graph pooling and *graph rewiring* have been proposed in the literature as solutions to address these limitations [14]. Given that the main infrastructure for message passing in MPNNs are the edges in the graph, and given that many of these edges might be noisy or inadequate for the downstream task [21], graph rewiring aims to identify such edges and edit them.

Many graph rewiring methods rely on edge sampling strategies: first, the edges are assigned new weights according to a *relevance function* and then they are re-sampled according to the new weights to retain the most relevant edges (i.e. those with larger weights). Edge relevance might be computed in different ways, including randomly [22], based on similarity [23] or on the edge's curvature [20].

Due to the diversity of possible graphs and tasks to be performed with those graphs, optimal graph rewiring should include a *variety of strategies* that are suited not only to the task at hand but also to the nature and structure of the graph.

**Motivation.** State-of-the-art edge sampling strategies have three significant **limitations**. First, most of the proposed methods **fail to preserve the global topology of the graph**. Second, most graph rewiring methods are neither **differentiable** nor **inductive** [20]. Third, relevance functions that depend on a diffusion measure (typically in the spectral domain) are **not parameter-free**, which adds a layer of complexity in the models. In this paper, we address these three limitations.

**Contributions and Outline.** The main contribution of this work is to propose a theoretical framework called DIFFWIRE for graph rewiring in GNNs that is principled, differentiable, inductive, and parameter-free by leveraging the Lovász bound [15] given by Eq. 1. This bound is a mathematical expression of the relationship between the *commute times* (*effective resistance distance*) and the network's *spectral gap*. Given an unseen test graph, DIFFWIRE predicts the optimal graph structure for the task at hand without any parameter tuning. Given the recently reported connection between commute times and curvature [24], and between curvature and the spectral gap [20], the proposed framework provides a unified theory linking these concepts. Our aim is to leverage diffusion and curvature theories to propose a new approach for graph rewiring that preserves the graph's structure.

We first propose using the CT as a relevance function for edge re-weighting. Moreover, we develop a differentiable, parameter-free layer in the GNN (CT-LAYER) to learn the CT. Second, we propose an alternative graph rewiring approach by adding a layer in the network (GAP-LAYER) that optimizes the spectral gap according to the nature of the network and the task at hand. Finally, we empirically validate the proposed layers with state-of-the-art benchmark datasets in a graph classification task. We test our approach on a graph classification task to emphasize the inductive nature of DIFFWIRE: the layers in the GNN (CT-LAYER or GAP-LAYER) are trained to predict the CTs embedding or minimize the spectral gap for unseen graphs, respectively. This approach gives a great advantage when compared to SoTA methods that require optimizing the parameters of the models for each graph. CT-LAYER and GAP-LAYER learn the weights during training to predict the optimal changes in the

topology of any unseen graph in test time. Finally, we also perform preliminary node classification experiments in heterophilic and homophilic graphs using CT-LAYER.

The paper is organized as follows: Section 2 provides a summary of the most relevant related literature. Our core technical contribution is described in Section 3, followed by our experimental evaluation and discussion in Section 4. Finally, Section 5 is devoted to conclusions and an outline of our future lines of research.

## 2   Related Work

In this section we provide an overview of the most relevant works that have been proposed in the literature to tackle the challenges of over-smoothing, over-squashing and under-reaching in MPNNs by means of graph rewiring and pooling.

**Graph Rewiring in MPNNs.** *Rewiring* is a process of changing the graph's structure to control the information flow and hence improve the ability of the network to perform the task at hand (e.g. node or graph classification, link prediction...). Several approaches have been proposed in the literature for graph rewiring, such as connectivity diffusion [25] or evolution [20], adding new bridge-nodes [26] and multi-hop filters [27], and neighborhood [12], node [28] and edge [22] sampling.

Edge sampling methods sample the graph's edges based on their weights or relevance, which might be computed in different ways. Rong et al. [22] show that randomly dropping edges during training improves the performance of GNNs. Klicpera et al. [25], define edge relevance according to the coefficients of a parameterized diffusion process over the graph and then edges are selected using a threshold rule. For Kazi et al. [23], edge relevance is given by the similarity between the nodes' attributes. In addition, a reinforcement learning process rewards edges leading to a correct classification and penalizes the rest.

Edge sampling-based rewiring has been proposed to tackle over-smoothing and over-squashing in MPNNs. Over-smoothing may be relieved by removing inter-class edges [29]. However, this strategy is only valid when the graph is homophilic, i.e. connected nodes tend to share similar attributes. Otherwise, removing these edges could lead to over-squashing [20] if their removal obstructs the message passing between distant nodes belonging to the same class (heterophily). Increasing the size of the bottlenecks of the graph via rewiring has been shown to improve node classification performance in heterophilic graphs, but not in homophilic graphs [20]. Recently, Topping et al. [20] propose an edge relevance function given by the edge curvature to mitigate over-squashing. They identify the bottleneck of the graph by computing the Ricci curvature of the edges. Next, they remove edges with high curvature and add edges around minimal curvature edges.

**Graph Structure Learning (GSL).** GSL methods [30] aim to learn an optimized graph structure and its corresponding representations *at the same time*. DIFFWIRE could be seen from the perspective of GSL: CT-LAYER, as a metric-based, neural approach, and GAP-LAYER, as a direct-neural approach to optimize the structure of the graph to the task at hand.

**Graph Pooling.** *Pooling* layers simplify the original graph by compressing it into a smaller graph or a vector via pooling operators, which range from simple [31] to more sophisticated approaches, such as DiffPool [32] and MinCut pool [33]. Although graph pooling methods do not consider the edge representations, there is a clear relationship between pooling methods and rewiring since both of them try to quantify the flow of information through the graph's bottleneck.

**Positional Encodings (PEs)** A Positional Encoding is a feature that describes the global or local position of the nodes in the graph. These features are related to random walk measures and the Laplacian's eigenvectors [34]. Commute Times embeddings (CTEs) may be considered an expressive form of PEs due to their spectral properties, i.e. their relation with the shortest path, the spectral gap or Cheeger constant. Velingker et al. [35] recently proposed use the CTEs as PE or commute times (CT) as edge feature. They pre-compute the CTEs and CT and add it as node or edge features to improve the structural expressiveness of the GNN. PEs are typically pre-computed and then used to build more expressive graph architectures, either by concatenating them to the node features or by building transformer models [36, 37]. Our work is related to PEs as CT-LAYER learns the original PEs from the input $\mathbf{X}$ and the adjacency matrix $\mathbf{A}$ instead of pre-computing and potentially modifying them, as previous works do [35–38]. Thus, CT-LAYER may be seen as a method to automatically learn the PEs for graph rewiring.

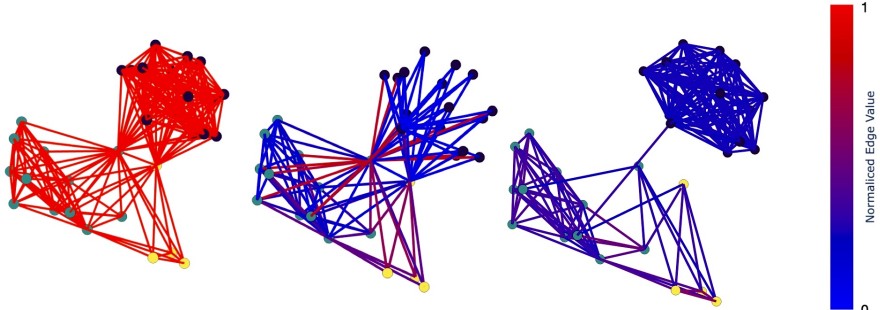

**Figure 1:** DIFFWIRE. Left: Original graph from COLLAB (test set). Center: Rewired graph after CT-LAYER. Right: Rewired graph after GAP-LAYER. Colors indicate the strength of the edges.

## 3 Proposed Approach: DIFFWIRE for Inductive Graph Rewiring

DIFFWIRE provides a unified theory for graph rewiring by proposing two new, complementary layers in MPNNs: first, CT-LAYER, a layer that learns the commute times and uses them as a relevance function for edge re-weighting; and second, GAP-LAYER, a layer to optimize the spectral gap, depending on the nature of the network and the task at hand.

In this section, we present the theoretical foundations for the definitions of CT-LAYER and GAP-LAYER. First, we introduce the bound that our approach is based on: The Lovász bound. Table 3 in A.1 summarizes the notation used in the paper.

### 3.1 The Lovász Bound

The Lovász bound, given by Eq. 1, was derived by Lovász in [15] as a means of linking the spectrum governing a random walk in an undirected graph $G = (V, E)$ with the *hitting time* $H_{uv}$ between any two nodes $u$ and $v$ of the graph. $H_{uv}$ is the expected number of steps needed to reach (or hit) $v$ from $u$; $H_{vu}$ is defined analogously. The sum of both hitting times between the two nodes, $v$ and $u$, is the *commute time* $CT_{uv} = H_{uv} + H_{vu}$. Thus, $CT_{uv}$ is the expected number of steps needed to hit $v$ from $u$ and go back to $u$. According to the Lovász bound:

$$\left| \frac{1}{vol(G)} CT_{uv} - \left( \frac{1}{d_u} + \frac{1}{d_v} \right) \right| \leq \frac{1}{\lambda_2'} \frac{2}{d_{min}} \tag{1}$$

where $\lambda_2' \geq 0$ is the *spectral gap*, i.e. the first non-zero eigenvalue of $\mathcal{L} = \mathbf{I} - \mathbf{D}^{-1/2} \mathbf{A} \mathbf{D}^{-1/2}$ (normalized Laplacian [39], where $\mathbf{D}$ is the degree matrix and $\mathbf{A}$, the adjacency matrix); $vol(G)$ is the volume of the graph (sum of degrees); $d_u$ and $d_v$ are the degrees of nodes $u$ and $v$, respectively; and $d_{min}$ is the minimum degree of the graph.

The term $CT_{uv}/vol(G)$ in Eq. 1 is referred to as the *effective resistance*, $R_{uv}$, between nodes $u$ and $v$. The bound states that the effective resistance between two nodes in the graph converges to or diverges from $(1/d_u + 1/d_v)$, depending on whether the graph's spectral gap diverges from or tends to zero. The larger the spectral gap, the closer $CT_{uv}/vol(G)$ will be to $\frac{1}{d_u} + \frac{1}{d_v}$ and hence the less informative the commute times will be.

We propose two novel GNNs layers based on each side of the inequality in Eq. 1: CT-LAYER, focuses on the left-hand side, and GAP-LAYER, on the right-hand side. The use of each layer depends on the nature of the network and the task at hand. In a graph classification task (our focus), CT-LAYER is expected to yield good results when the graph's spectral gap is small; conversely, GAP-LAYER would be the layer of choice in graphs with large spectral gap.

The Lovász bound was later refined by von Luxburg et al. [40]. App. A.2.2 presents this bound along with its relationship with $R_{uv}$ as a global measure of node similarity. Once we have defined both sides of the Lovász bound, we proceed to describe their implications for graph rewiring.

## 3.2 CT-LAYER: Commute Times for Graph Rewiring

We focus first on the left-hand side of the Lovász bound which concerns the effective resistances $CT_{uv}/vol(G) = R_{uv}$ (or commute times)[1] between any two nodes in the graph.

**Spectral Sparsification leads to Commute Times.** Graph sparsification in undirected graphs may be formulated as finding a graph $H = (V, E')$ that is *spectrally similar* to the original graph $G = (V, E)$ with $E' \subset E$. Thus, the spectra of their Laplacians, $\mathbf{L}_G$ and $\mathbf{L}_H$ should be similar.

**Theorem 1** (Spielman and Srivastava [41]). *Let* Sparsify*(G, q) –> G' be a sampling algorithm of graph $G = (V, E)$, where edges $e \in E$ are sampled with probability $q \propto R_e$ (proportional to the effective resistance). For $n = |V|$ sufficiently large and $1/\sqrt{n} < \epsilon \leq 1$, $O(n \log n/\epsilon^2)$ samples are needed to satisfy $\forall \mathbf{x} \in \mathbb{R}^n$: $(1 - \epsilon)\mathbf{x}^T\mathbf{L}_G\mathbf{x} \leq \mathbf{x}^T\mathbf{L}_{G'}\mathbf{x} \leq (1 + \epsilon)\mathbf{x}^T\mathbf{L}_G\mathbf{x}$, with probability $\geq 1/2$.*

The above theorem has a simple explanation in terms of Dirichlet energies, $\mathcal{E}(\mathbf{x})$. The Laplacian $\mathbf{L} = \mathbf{D} - \mathbf{A} \succeq 0$, i.e. it is positive semi-definite (all its eigenvalues are non-negative). Then, if we consider $\mathbf{x} : V \rightarrow \mathbb{R}$ as a real-valued function of the $n$ nodes of $G = (V, E)$, we have that $\mathcal{E}(\mathbf{x}) := \mathbf{x}^T\mathbf{L}_G\mathbf{x} = \sum_{e=(u,v)\in E}(\mathbf{x}_u - \mathbf{x}_v)^2 \geq 0$ for any $\mathbf{x}$. In particular, the eigenvectors $\mathbf{f} := \{\mathbf{f}_i : \mathbf{L}\mathbf{f}_i = \lambda_i\mathbf{f}_i\}$ are the set of special functions that minimize the energies $\mathcal{E}(\mathbf{f}_i)$, i.e. they are the mutually orthogonal and normalized functions with the minimal variabilities achievable by the topology of $G$. Therefore, any minimal variability of $G'$ is bounded by $(1 \pm \epsilon)$ times that of $G$ if we sample enough edges with probability $q \propto R_e$. In addition, $\lambda_i = \frac{\mathcal{E}(\mathbf{f}_i)}{\mathbf{f}_i^T\mathbf{f}_i}$.

This first result implies that edge sampling based on commute times is a principled way to rewire a graph while preserving its original structure and it is bounded by the Dirichlet energies. Next, we present what a commute times embedding is and how it can be spectrally computed.

**Commute Times Embedding (CTE).** The choice of effective resistances in Theorem 1 is explained by the fact that $R_{uv}$ can be computed from $R_{uv} = (\mathbf{e}_u - \mathbf{e}_v)^T\mathbf{L}^+(\mathbf{e}_u - \mathbf{e}_v)$, where $\mathbf{e}_u$ is the unit vector with a unit value at $u$ and zero elsewhere. $\mathbf{L}^+ = \sum_{i\geq 2} \lambda_i^{-1}\mathbf{f}_i\mathbf{f}_i^T$, where $\mathbf{f}_i, \lambda_i$ are the eigenvectors and eigenvalues of $\mathbf{L}$, is the pseudo-inverse or Green's function of $G = (V, E)$ if it is connected. The Green's function leads to envision $R_{uv}$ (and therefore $CT_{uv}$) as *metrics* relating pairs of nodes of $G$. As a result, the CTE will preserve the commute times distance in a Euclidean space. Note that this latent space of the nodes can not only be described spectrally but also in a *parameter free*-manner, which is not the case for other spectral embeddings, such as heat kernel or diffusion maps as they rely on a time parameter $t$. More precisely, the embedding matrix $\mathbf{Z}$ whose columns contain the nodes' commute times embeddings is spectrally given by:

$$\mathbf{Z} := \sqrt{vol(G)}\Lambda^{-1/2}\mathbf{F}^T = \sqrt{vol(G)}\Lambda'^{-1/2}\mathbf{G}^T\mathbf{D}^{-1/2} \tag{2}$$

where $\Lambda$ is the diagonal matrix of the unnormalized Laplacian $\mathbf{L}$ eigenvalues and $\mathbf{F}$ is the matrix of their associated eigenvectors. Similarly, $\Lambda'$ contains the eigenvalues of the normalized Laplacian $\mathcal{L}$ and $\mathbf{G}$ the eigenvectors. We have $\mathbf{F} = \mathbf{G}\mathbf{D}^{-1/2}$ or $\mathbf{f}_i = \mathbf{g}_i\mathbf{D}^{-1/2}$, where $\mathbf{D}$ is the degree matrix.

Finally, the commute times are given by the Euclidean distances between the embeddings $CT_{uv} = \|\mathbf{z}_u - \mathbf{z}_v\|^2$. The spectral calculation of commute times distances is given by:

$$R_{uv} = \frac{CT_{uv}}{vol(G)} = \frac{\|\mathbf{z}_u - \mathbf{z}_v\|^2}{vol(G)} = \sum_{i=2}^{n} \frac{1}{\lambda_i} \left(\mathbf{f}_i(u) - \mathbf{f}_i(v)\right)^2 = \sum_{i=2}^{n} \frac{1}{\lambda_i'} \left(\frac{\mathbf{g}_i(u)}{\sqrt{d_u}} - \frac{\mathbf{g}_i(v)}{\sqrt{d_v}}\right)^2 \tag{3}$$

**Commute Times as an Optimization Problem.** In this section, we demonstrate how the CTs may be computed as an optimization problem by means of a differentiable layer in a GNN. Constraining neighboring nodes to have a similar embedding leads to

$$\mathbf{Z} = \arg \min_{\mathbf{Z}^T\mathbf{Z}=\mathbf{I}} \frac{\sum_{u,v} \|\mathbf{z}_u - \mathbf{z}_v\|^2\mathbf{A}_{uv}}{\sum_{u,v} \mathbf{Z}_{uv}^2 d_u} = \frac{\sum_{(u,v)\in E} \|\mathbf{z}_u - \mathbf{z}_v\|^2}{\sum_{u,v} \mathbf{Z}_{uv}^2 d_u} = \frac{Tr[\mathbf{Z}^T\mathbf{L}\mathbf{Z}]}{Tr[\mathbf{Z}^T\mathbf{D}\mathbf{Z}]} \,, \tag{4}$$

which reveals that CTs embeddings result from a Laplacian regularization down-weighted by the degree. As a result, *frontier* nodes or hubs –i.e. nodes with inter-community edges– which tend to

---

[1]We use commute times and effective resistances interchangeably as per their use in the literature

have larger degrees than those lying inside their respective communities will be embedded far away from their neighbors, increasing the *distance* between communities. Note that the above *quotient of traces* formulation is easily differentiable and different from $Tr[\frac{\mathbf{Z}^T\mathbf{LZ}}{\mathbf{Z}^T\mathbf{DZ}}]$ proposed in [42].

With the above elements we define CT-LAYER, the first rewiring layer proposed in this paper. See Figure 2 for a graphical representation of the layer.

**Definition 1** (CT-Layer). *Given the matrix $\mathbf{X}_{n\times F}$ encoding the features of the nodes after any message passing (MP) layer, $\mathbf{Z}_{n\times O(n)} = \tanh(\mathrm{MLP}(\mathbf{X}))$ learns the association $\mathbf{X} \to \mathbf{Z}$ while $\mathbf{Z}$ is optimized according to the loss $L_{CT} = \frac{Tr[\mathbf{Z}^T\mathbf{LZ}]}{Tr[\mathbf{Z}^T\mathbf{DZ}]} + \left\| \frac{\mathbf{Z}^T\mathbf{Z}}{\|\mathbf{Z}^T\mathbf{Z}\|_F} - \mathbf{I}_n \right\|_F$. This results in the following resistance diffusion $\mathbf{T}^{CT} = \mathbf{R}(\mathbf{Z}) \odot \mathbf{A}$, i.e. the Hadamard product between the resistance distance and the adjacency matrix, providing as input to the subsequent MP layer a learnt convolution matrix. We set $\mathbf{R}(\mathbf{Z})$ to the pairwise Euclidean distances of the node embeddings in $\mathbf{Z}$ divided by $vol(G)$.*

Thus, CT-LAYER learns the CTs and rewires an input graph according to them: the edges with maximal resistance will tend to be the most important edges so as to preserve the topology of the graph.

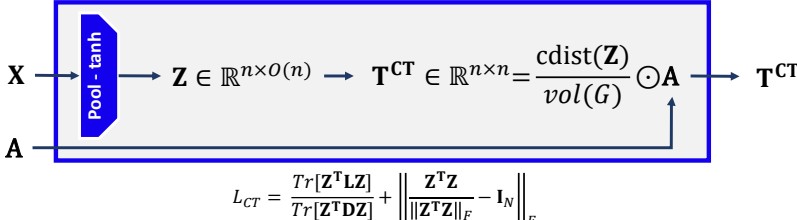

$$L_{CT} = \frac{Tr[\mathbf{Z^TLZ}]}{Tr[\mathbf{Z^TDZ}]} + \left\| \frac{\mathbf{Z^TZ}}{\|\mathbf{Z^TZ}\|_F} - \mathbf{I}_N \right\|_F$$

**Figure 2:** Detailed depiction of CT-LAYER, where `cdist` refers to the matrix of pairwise Euclidean distances between the node embeddings in $\mathbf{Z}$.

Below, we present the relationship between the CTs and the graph's bottleneck and curvature.

$\mathbf{T}^{CT}$ **and Graph Bottlenecks.** Beyond the principled sparsification of $\mathbf{T}^{CT}$ (Theorem 1), this layer rewires the graph $G = (E, V)$ in such a way that edges with maximal resistance will tend to be the most critical to preserve the topology of the graph. More precisely, although $\sum_{e\in E} R_e = n - 1$, the bulk of the resistance distribution will be located at graph bottlenecks, if they exist. Otherwise, their magnitude is upper-bounded and the distribution becomes more uniform.

Graph bottlenecks are controlled by the *graph's conductance* or Cheeger constant, $h_G = min_{S\subseteq V} h_S$, where: $h_S = \frac{|\partial S|}{\min(vol(S),vol(\bar{S}))}$, $\partial S = \{e = (u,v) : u \in S, v \in \bar{S}\}$ and $vol(S) = \sum_{u\in S} d_u$.

The interplay between the graph's conductance and effective resistances is given by:
**Theorem 2** (Alev et al. [43]). *Given a graph $G = (V, E)$, a subset $S \subseteq V$ with $vol(S) \leq vol(G)/2$,*

$$h_S \geq \frac{c}{vol(S)^{1/2-\epsilon}} \iff |\partial S| \geq c \cdot vol(S)^{1/2-\epsilon}, \tag{5}$$

*for some constant $c$ and $\epsilon \in [0, 1/2]$. Then, $R_{uv} \leq \left(\frac{1}{d_u^{2\epsilon}} + \frac{1}{d_v^{2\epsilon}}\right) \cdot \frac{1}{\epsilon\cdot c^2}$ for any pair $u, v$.*

According to this theorem, the larger the graph's bottleneck, the tighter the bound on $R_{uv}$ are. Moreover, $\max(R_{uv}) \leq 1/h_S^2$, i.e., the resistance is bounded by the square of the bottleneck.

This bound partially explains the rewiring of the graph in Figure 1-center. As seen in the Figure 1-center, rewiring using CT-LAYER sparsifies the graph and assigns larger weights to the edges located in the graph's bottleneck. The interplay between Theorem 2 and Theorem 1 is described in App. A.1.

Recent work has proposed using curvature for graph rewiring. We outline below the relationship between CTs and curvature.

**Effective Resistances and Curvature.** Topping et al. [20] propose an approach for graph rewiring, where the relevance function is given by the Ricci curvature. However, this measure is non-differentiable. More recent definitions of curvature [24] have been formulated based on resistance

distances that would be differentiable using our approach. The resistance curvature of an edge $e = (u,v)$ is $\kappa_{uv} := 2(p_u + p_v)/R_{uv}$ where $p_u := 1 - \frac{1}{2}\sum_{u\sim w} R_{uv}$ is the node's curvature. Relevant properties of the edge resistance curvature are discussed in App. A.1.3, along with a related Theorem proposed in Devriendt and Lambiotte [24].

### 3.3 GAP-LAYER: Spectral Gap Optimization for Graph Rewiring

The right-hand side of the Lovász bound in Eq. 1 relies on the graph's spectral gap $\lambda'_2$, such that the larger the spectral gap, the closer the commute times would be to their non-informative regime. Note that the spectral gap is typically large in commonly observed graphs –such as communities in social networks which may be bridged by many edges [44]– and, hence, in these cases it would be desirable to rewire the adjacency matrix $\mathbf{A}$ so that $\lambda'_2$ is minimized.

In this section, we explain how to rewire the graph's adjacency matrix A to minimize the spectral gap. We propose using the gradient of $\lambda_2$ wrt each component of $\tilde{\mathbf{A}}$. Then, we can compute these gradient either using Laplacians ($\mathbf{L}$, with Fiedler $\lambda_2$) or normalized Laplacians ($\mathcal{L}$, with Fiedler $\lambda'_2$). We also present an approximation of the Fiedler vectors needed to compute those gradients, and propose computing them as a GNN Layer called the GAP-LAYER. A detailed schematic of GAP-LAYER is shown in Figure 3.

**Rewiring using a Ratio-cut (Rcut) Approximation.** We propose to rewire the adjacency matrix, $\mathbf{A}$, so that $\lambda_2$ is minimized. We consider a matrix $\tilde{\mathbf{A}}$ close to $\mathbf{A}$ that satisfies $\tilde{\mathbf{L}}\mathbf{f}_2 = \lambda_2\mathbf{f_2}$, where $\mathbf{f}_2$ is the solution to the ratio-cut relaxation [45]. Following [46], the gradient of $\lambda_2$ wrt each component of $\tilde{\mathbf{A}}$ is given by

$$\nabla_{\tilde{\mathbf{A}}}\lambda_2 := Tr\left[(\nabla_{\tilde{\mathbf{L}}}\lambda_2)^T \cdot \nabla_{\tilde{\mathbf{A}}}\tilde{\mathbf{L}}\right] = \text{diag}(\mathbf{f}_2\mathbf{f}_2^T)\mathbf{1}\mathbf{1}^T - \mathbf{f}_2\mathbf{f}_2^T \tag{6}$$

where $\mathbf{1}$ is the vector of $n$ ones; and $[\nabla_{\tilde{\mathbf{A}}}\lambda_2]_{ij}$ is the gradient of $\lambda_2$ wrt $\tilde{\mathbf{A}}_{uv}$. The driving force of this gradient relies on the correlation $\mathbf{f}_2\mathbf{f}_2^T$. Using this gradient to minimize $\lambda_2$ results in breaking the graph's bottleneck while preserving simultaneously the inter-cluster structure. We delve into this matter in App. A.2.

**Rewiring using a Normalized-cut (Ncut) Approximation.** Similarly, considering now $\lambda'_2$ for rewiring leads to

$$\nabla_{\tilde{\mathbf{A}}}\lambda'_2 := Tr\left[(\nabla_{\tilde{\mathcal{L}}}\lambda_2)^T \cdot \nabla_{\tilde{\mathbf{A}}}\tilde{\mathcal{L}}\right] =$$
$$\mathbf{d}'\left\{\mathbf{g}_2^T\tilde{\mathbf{A}}^T\tilde{\mathbf{D}}^{-1/2}\mathbf{g}_2\right\}\mathbf{1}^T + \mathbf{d}'\left\{\mathbf{g}_2^T\tilde{\mathbf{A}}\tilde{\mathbf{D}}^{-1/2}\mathbf{g}_2\right\}\mathbf{1}^T \quad + \quad \tilde{\mathbf{D}}^{-1/2}\mathbf{g}_2\mathbf{g}_2^T\tilde{\mathbf{D}}^{-1/2} \tag{7}$$

where $\mathbf{d}'$ is a $n \times 1$ vector including derivatives of degree wrt adjacency and related terms. This gradient relies on the Fiedler vector $\mathbf{g}_2$ (the solution to the normalized-cut relaxation), and on the incoming and outgoing one-hop random walks. This approximation breaks the bottleneck while preserving the global topology of the graph (Figure 1-left). Proof and details are included in App. A.2.

We present next an approximation of the Fiedler vector, followed by a proposed new layer in the GNN called the GAP-LAYER to learn how to minimize the spectral gap of the graph.

**Approximating the Fiedler vector.** Given that $\mathbf{g}_2 = \tilde{\mathbf{D}}^{1/2}\mathbf{f}_2$, we can obtain the normalized-cut gradient in terms of $\mathbf{f}_2$. From [17] we have that

$$\mathbf{f}_2(u) = \begin{cases} +1/\sqrt{n} & \text{if } u \text{ belongs to the first cluster} \\ -1/\sqrt{n} & \text{if } u \text{ belongs to the second cluster} \end{cases} + O\left(\frac{\log n}{n}\right) \tag{8}$$

**Definition 2** (GAP-Layer). *Given the matrix $\mathbf{X}_{n\times F}$ encoding the features of the nodes after any message passing (MP) layer, $\mathbf{S}_{n\times 2} = Softmax(\text{MLP}(\mathbf{X}))$ learns the association $\mathbf{X} \to \mathbf{S}$ while $\mathbf{S}$ is optimized according to the loss $L_{Cut} = -\frac{Tr[\mathbf{S}^T\mathbf{A}\mathbf{S}]}{Tr[\mathbf{S}^T\mathbf{D}\mathbf{S}]} + \left\|\frac{\mathbf{S}^T\mathbf{S}}{\|\mathbf{S}^T\mathbf{S}\|_F} - \frac{\mathbf{I}_n}{\sqrt{2}}\right\|_F$. Then the Fiedler vector $\mathbf{f}_2$ is approximated by applying a softmaxed version of Eq. 8 and considering the loss $L_{Fiedler} = \|\tilde{\mathbf{A}} - \mathbf{A}\|_F + \alpha(\lambda_2^*)^2$, where $\lambda_2^* = \lambda_2$ if we use the ratio-cut approximation (and gradient) and $\lambda_2^* = \lambda'_2$ if we use the normalized-cut approximation and gradient. This returns $\tilde{\mathbf{A}}$ and the GAP diffusion $\mathbf{T}^{GAP} = \tilde{\mathbf{A}}(\mathbf{S}) \odot \mathbf{A}$ results from minimizing $L_{GAP} := L_{Cut} + L_{Fiedler}$.*

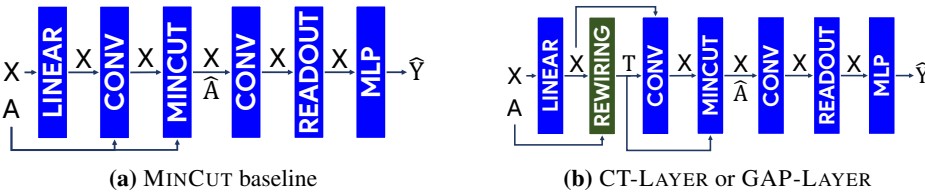

$$L_{cut} = \frac{Tr[\mathbf{S^T LS}]}{Tr[\mathbf{S^T DS}]} + \left\| \frac{\mathbf{S^T S}}{\|\mathbf{S^T S}\|_F} - \frac{\mathbf{I}_N}{\sqrt{2}} \right\|_F \qquad L_{fiedler} = \left\| \widetilde{\mathbf{A}} - \mathbf{A} \right\|_F + \alpha(\lambda_2)^2 \\ \nabla_{\widetilde{\mathbf{A}}}\lambda_2 = \left[ 2(\widetilde{\mathbf{A}} - \mathbf{A}) + (\mathrm{diag}(\mathbf{f}_2 \mathbf{f}_2^T)\mathbf{1}\mathbf{1}^T - \mathbf{f}_2 \mathbf{f}_2^T) \times \lambda_2 \right]$$

**Figure 3:** GAP-LAYER (Rcut). For GAP-LAYER (Ncut), substitute $\nabla_{\widetilde{\mathbf{A}}}\lambda_2$ by Eq. 7

## 4 Experiments and Discussion

### 4.1 Graph Classification

In this section, we study the properties and performance of CT-LAYER and GAP-LAYER in a graph classification task with several benchmark datasets. To illustrate the merits of our approach, we compare CT-LAYER and GAP-LAYER with 3 state-of-the-art diffusion and curvature-based graph rewiring methods. Note that the aim of the evaluation is to shed light on the properties of both layers and illustrate their inductive performance, not to perform a benchmark comparison with all previously proposed graph rewiring methods.

**(a)** MINCUT baseline         **(b)** CT-LAYER or GAP-LAYER

**Figure 4:** GNN models used in the experiments. Left: MinCut Baseline model. Right: CT-LAYER or GAP-LAYER models, depending on what method is used for rewiring.

**Baselines:**. The first baseline architecture is based on **MINCUT Pool** [33] and it is shown in Figure 4a. It is the base GNN that we use for graph classification without rewiring. MINCUT Pool layer learns $(\mathbf{A}_{n\times n}, \mathbf{X}_{n\times F}) \to (\mathbf{A}'_{k\times k}, \mathbf{X}_{k\times F})$, being $k < n$ the new number of node clusters. The first baseline strategy using graph rewiring is $k$-NN graphs [47], where weights of the edges are computed based on feature similarity. The next two baselines are graph rewiring methods that belong to the same family of methods as DIFFWIRE, i.e. methods based on diffusion and curvature, namely **DIGL** (PPR) [25] and **SDRF** [20]. DIGL is a diffusion-based preprocessing method within the family of metric-based GSL approaches. We set the teleporting probability $\alpha = 0.001$ and $\epsilon$ is set to keep the same average degree for each graph. Once preprocessed with DIGL, the graphs are provided as input to the MinCut Pool (Baseline1) arquitecture. The third baseline model is SDRF, which performs curvature-based rewiring. SDRF is also a preprocessing method which has 3 parameters that are highly graph-dependent. We set these parameters to $\tau = 20$ and $C^+ = 0$ for all experiments as per [20]. The number of iterations is estimated dynamically according to $0.7 * |V|$ for each graph.

Both DIGL and SDRF aim to preserve the global topology of the graph but require optimizing their parameters for each input graph via hyper-parameter search. In a graph classification task, this search is $O(n^3)$ per graph. Details about the parameter tuning in these methods can be found in App. A.3.3.

To shed light on the performance and properties of CT-LAYER and GAP-LAYER, we add the corresponding layer in between $\text{Linear}(\mathbf{X}) \xrightarrow{*} \text{Conv1}(\mathbf{A}, \mathbf{X})$. We build 3 different models: CT-LAYER, GAP-LAYER (Rcut), GAP-LAYER (Ncut), depending on the layer used. For CT-LAYER, we learn $\mathbf{T}^{CT}$ which is used as a convolution matrix afterwards. For GAP-LAYER, we learn $\mathbf{T}^{GAP}$ either using the Rcut or the Ncut approximations. A schematic of the architectures is shown in Figure 4b and in App. A.3.2.

As shown in Table 1, we use in our experiments common benchmark datasets for graph classification. We select datasets both with features and featureless, in which case we use the degree as the node features. These datasets are diverse regarding the topology of their networks: REDDIT-B, IMDB-B

**Table 1:** Experimental results on common graph classification benchmarks. **Red** denotes the best model row-wise and **Blue** marks the runner-up. '*' means degree as node feature.

|  | MinCutPool | $k$-NN | DIGL | SDRF | CT-Layer | GAP-Layer (R) | GAP-Layer (N) |
|---|---|---|---|---|---|---|---|
| REDDIT-B* | $66.53{\pm}4.4$ | $64.40{\pm}3.8$ | $76.02{\pm}4.3$ | $65.3{\pm}7.7$ | **$78.45{\pm}4.5$** | **$77.63{\pm}4.9$** | $76.00{\pm}5.3$ |
| IMDB-B* | $60.75{\pm}7.0$ | $55.20{\pm}4.3$ | $59.35{\pm}7.7$ | $59.2{\pm}6.9$ | **$69.84{\pm}4.6$** | **$69.93{\pm}3.3$** | $68.80{\pm}3.1$ |
| COLLAB* | $58.00{\pm}6.2$ | $58.33{\pm}11$ | $57.51{\pm}5.9$ | $56.60{\pm}10$ | **$69.87{\pm}2.4$** | $64.47{\pm}4.0$ | **$65.89{\pm}4.9$** |
| MUTAG | $84.21{\pm}6.3$ | **$87.58{\pm}4.1$** | $85.00{\pm}5.6$ | $82.4{\pm}6.8$ | **$87.58{\pm}4.4$** | $86.90{\pm}4.0$ | $86.90{\pm}4.0$ |
| PROTEINS | $74.84{\pm}2.3$ | **$76.76{\pm}2.5$** | $74.49{\pm}2.8$ | $74.4{\pm}2.7$ | **$75.38{\pm}2.9$** | $75.03{\pm}3.0$ | $75.34{\pm}2.1$ |
| SBM* | $53.00{\pm}9.9$ | $50.00{\pm}0.0$ | $56.93{\pm}12$ | $54.1{\pm}7.1$ | $81.40{\pm}11$ | **$90.80{\pm}7.0$** | **$92.26{\pm}2.9$** |
| Erdös-Rényi* | $81.86{\pm}6.2$ | $63.40{\pm}3.9$ | **$81.93{\pm}6.3$** | $73.6{\pm}9.1$ | $79.06{\pm}9.8$ | $79.26{\pm}10$ | **$82.26{\pm}3.2$** |

and COLLAB contain truncate scale-free graphs (social networks), whereas MUTAG and PROTEINS contain graphs from biology or chemistry. In addition, we use two synthetic datasets with 2 classes: Erdös-Rényi with $p_1 \in [0.3, 0.5]$ and $p_2 \in [0.4, 0.8]$ and Stochastic block model (SBM) with parameters $p_1 = 0.8$, $p_2 = 0.5$, $q_1 \in [0.1, 0.15]$ and $q_2 \in [0.01, 0.1]$. More details about the datasets in App. A.3.1. In addition, Table 1 reports average accuracies and standard deviation on 10 random data splits, using 85/15 stratified train-test split, training during 60 epochs and reporting the results of the last epoch for each random run. We use Pytorch Geometric [48] and the code is available in a public repository[2].

The experiments support our hypothesis that rewiring based on CT-Layer and GAP-Layer improves the performance of the baselines on graph classification. Since both layers are differentiable, they learn how to inductively rewire unseen graphs. The improvements are significant in graphs where social components arise (REDDITB, IMDBB, COLLAB), i.e. graphs with small world properties and power-law degree distributions with a topology based on hubs and authorities. These are graphs where bottlenecks arise easily and our approach is able to properly rewire the graphs. However, the improvements observed in planar or grid networks (MUTAG and PROTEINS) are more limited: the bottleneck does not seem to be critical for the graph classification task.

Moreover, CT-Layer and GAP-Layer perform better in graphs with featureless nodes than graphs with node features because it is able to leverage the information encoded in the topology of the graphs. Note that in attribute-based graphs, the weights of the attributes typically overwrite the graph's structure in the classification task, whereas in graphs without node features, the information is encoded in the graph's structure. Thus, $k$-NN rewiring outperforms every other rewiring method in graph classification where graphs has node features.

App. A.3.4 contains an in-depth analysis of the comparison between the spectral node CT embeddings (CTEs) given by Equation 2, and the learned node CTEs as predicted by CT-Layer. We find that the CTEs that are learned in CT-Layer are able to better preserve the original topology of the graph while shifting the distribution of the effective resistances of the edges towards an asymmetric distribution where few edges have very large weights and a majority of edges have low weights.

In addition, App. A.3.4 also includes the analysis of the graphs latent space of the readout layer produced by each model. Finally, we analyze the performance of the proposed layers in graphs with different structural properties in App. A.3.6. We analyze the correlation between accuracy, the graph's assortativity, and the graph's bottleneck ($\lambda_2$).

**CT-Layer vs GAP-Layer.** The datasets explored in this paper are characterized by mild bottlenecks from the perspective of the Lovász bound. For completion, we have included two synthetic datasets (Stochastic Block Model and Erdös-Rényi) where the Lovász bound is very restrictive. As a result, CT-Layer is outperformed by GAP-Layer in SBM. Note that the results on the synthetic datasets suffer from large variability. As a general rule of thumb, the smaller the graph's bottleneck (defined as the ratio between the number of inter-community edges and the number of intra-community edges), the more useful the CT-Layer is because the rewired graph will be sparsified in the communities but will preserve the edges in the gap. Conversely, the larger the bottleneck, the more useful the GAP-Layer is.

---

[2] https://github.com/AdrianArnaiz/DiffWire

## 4.2 Node Classification using CT-LAYER

CT-LAYER and GAP-LAYER are mainly designed to perform graph classification tasks. However, we identify two potential areas to apply CT-LAYER for node classification.

First, the new $\mathbf{T^{CT}}$ diffusion matrix learned by CT-LAYER gives more importance to edges that connect different communities, i.e., edges that connect distant nodes in the graph. This behaviour of CT-LAYER is aligned to solve long-range and *heterophilic* node classification tasks using fewer number of layers and thus avoiding under-reaching, over-smoothing and over-squashing.

Second, there is an increasingly interest in the community in using PEs in the nodes to develope more expressive GNN. PEs tend to help in node classification in *homophilic* graphs, as nearby nodes will be assigned similar PEs. However, the main limitation is that PEs are usually pre-computed before the GNN training due to their high computational cost. CT-LAYER provides a solution to this problem, as it *learns* to predict the commute times embedding ($\mathbf{Z}$) of a given graph (see Figure 2 and definition 1). Hence, CT-LAYER is able to learn and predict PEs from $\mathbf{X}$ and $\mathbf{A}$ inside a GNN without needing to pre-compute them.

We empirically validate CT-LAYER in a node classification task on benchmark homophilic (Cora, Pubmed and Citeseer) and heterophilic (Cornell, Actor and Wisconsin) graphs. The results are depicted in Table 2 comparing three models: (1) the baseline model consists of a 1-layer-GCN; (2) *model 1* is a 1-layer-GCN where the CTEs are concatenated to the node features as PEs ($\mathbf{X} \parallel \mathbf{Z}$); (3) Finally, *model 2* is a 1-layer-GCN where $\mathbf{T^{CT}}$ is used as a diffusion matrix ($\mathbf{A} = \mathbf{T^{CT}}$). More details can be found in App. A.3.5.

As seen in the Table, the proposed models outperform the baseline GCN model: using CTEs as features (model 1) yields competitive results in homophilic graphs whereas using $\mathbf{T^{CT}}$ as a matrix for message passing (model 2) performs well in heterophilic graphs. Note that in our experiments the CTEs are learned by CT-LAYER instead of being pre-computed. A promising direction of future work would be to explore how to combine these two approaches (model 1 and model 2) to leverage the best of each of the methods on a wide range of graphs for node classification tasks.

**Table 2:** Results in node classification

| Dataset | GCN (baseline) | *model 1:* $\mathbf{X} \parallel \mathbf{Z}$ | *model 2:* $\mathbf{A} = \mathbf{T^{CT}}$ | Homophily |
|---|---|---|---|---|
| Cora | $82.01_{\pm 0.8}$ | $\mathbf{83.66}_{\pm 0.6}$ | $67.96_{\pm 0.8}$ | *81.0%* |
| Pubmed | $81.61_{\pm 0.3}$ | $\mathbf{86.07}_{\pm 0.1}$ | $68.19_{\pm 0.7}$ | *80.0%* |
| Citeseer | $70.81_{\pm 0.5}$ | $\mathbf{72.26}_{\pm 0.5}$ | $66.71_{\pm 0.6}$ | *73.6%* |
| Cornell | $59.19_{\pm 3.5}$ | $58.02_{\pm 3.7}$ | $\mathbf{69.04}_{\pm 2.2}$ | *30.5%* |
| Actor | $29.59_{\pm 0.4}$ | $29.35_{\pm 0.4}$ | $\mathbf{31.98}_{\pm 0.3}$ | *21.9%* |
| Wisconsin | $68.05_{\pm 6.2}$ | $69.25_{\pm 5.1}$ | $\mathbf{79.05}_{\pm 2.1}$ | *19.6%* |

## 5 Conclusion and Future Work

In this paper, we have proposed DIFFWIRE, a unified framework for graph rewiring that links the two components of the Lovász bound: CTs and the spectral gap. We have presented two novel, fully differentiable and inductive rewiring layers: CT-LAYER and GAP-LAYER. We have empirically evaluated these layers on benchmark datasets for graph classification with competitive results when compared to SoTA baselines, specially in graphs where the the nodes have no attributes and have small-world properties. We have also performed preliminary experiments in a node classification task, showing that using the CT Embeddings and the CT distances benefit GNN architectures in homophilic and heterophilic graphs, respectively.

In future work, we plan to test the proposed approach in other graph-related tasks and intend to apply DIFFWIRE to large-scale graphs and real-world applications, particularly in social networks, which have unique topology, statistics and direct implications in society.

## 6 Acknowledgments

A. Arnaiz-Rodriguez and N. Oliver are supported by a nominal grant received at the ELLIS Unit Alicante Foundation from the Regional Government of Valencia in Spain (Convenio Singular signed with Generalitat Valenciana, Conselleria d'Innovació, Universitats, Ciència i Societat Digital, Dirección General para el Avance de la Sociedad Digital). A. Arnaiz-Rodriguez is also funded by a grant by the Banc Sabadell Foundation. F. Escolano is funded by the project RTI2018-096223-B-I00 of the Spanish Government.

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

# A    Appendix

In Appendix A we include a Table with the notation used in the paper and we provide an analysis of the diffusion and its relationship with curvature. In Appendix B, we study in detail GAP-LAYER and the implications of the proposed spectral gradients. Appendix C reports statistics and characteristics of the datasets used in the experimental section, provides more information about the experiments results, describes additional experimental results, and includes a summary of the computing infrastructure used in our experiments.

**Table 3:** Notation.

| Symbol | Description |
|---|---|
| $G = (V, E)$ | Graph = (Nodes, Edges) |
| $\mathbf{A}$ | Adjacency matrix: $\mathbf{A} \in \mathbb{R}^{n \times n}$ |
| $\mathbf{X}$ | Feature matrix: $\mathbf{X} \in \mathbb{R}^{n \times F}$ |
| $v$ | Node $v \in V$ or $u \in V$ |
| $e$ | Edge $e \in E$ |
| $x$ | Features of node $v$: $x \in X$ |
| $n$ | Number of nodes: $n = |V|$ |
| $F$ | Number of features |
| $\mathbf{D}$ | Degree diagonal matrix where $d_v$ in $D_{vv}$ |
| $d_v$ | Degree of node $v$ |
| $vol(G)$ | Sum of the degrees of the graph $vol(G) = Tr[D]$ |
| $\mathbf{L}$ | Laplacian: $\mathbf{L} = \mathbf{D} - \mathbf{A}$ |
| $\mathbf{B}$ | Signed edge-vertex incidence matrix |
| $\mathbf{b}_e$ | Incidence vector: Row vector of $\mathbf{B}$, with $\mathbf{b}_{e=(u,v)} = (\mathbf{e}_u - \mathbf{e}_v)$ |
| $\mathbf{v}_e$ | Projected incidence vector: $\mathbf{v}_e = \mathbf{L}^{+/2}\mathbf{b}_e$ |
| $\Gamma$ | Ratio $\Gamma = \frac{1+\epsilon}{1-\epsilon}$ |
| $\mathcal{E}$ | Dirichlet Energy wrt $\mathbf{L}$: $\mathcal{E}(\mathbf{x}) := \mathbf{x}^T \mathbf{L} \mathbf{x}$ |
| $\mathcal{L}$ | Normalized Laplacian: $\mathcal{L} = \mathbf{I} - \mathbf{D}^{-1/2}\mathbf{A}\mathbf{D}^{-1/2}$ |
| $\Lambda$ | Eigenvalue matrix of $\mathbf{L}$ |
| $\Lambda'$ | Eigenvalue matrix of $\mathcal{L}$ |
| $\lambda_i$ | $i$-th eigenvalue of $\mathbf{L}$ |
| $\lambda_2$ | Second eigenvalue of $\mathbf{L}$: Spectral gap |
| $\lambda'_i$ | $i$-th eigenvalue of $\mathcal{L}$ |
| $\lambda'_2$ | Second eigenvalue of $\mathcal{L}$: Spectral gap |
| $\mathbf{F}$ | Matrix of eigenvectors of $\mathbf{L}$ |
| $\mathbf{G}$ | Matrix of eigenvectors of $\mathcal{L}$ |
| $\mathbf{f}_i$ | $i$ eigenvector of $\mathbf{L}$ |
| $\mathbf{f}_2$ | Second eigenvector of $\mathbf{L}$: Fiedler vector |
| $\mathbf{g}_i$ | $i$ eigenvector of $\mathcal{L}$ |
| $\mathbf{g}_2$ | Second eigenvector of $\mathcal{L}$: Fiedler vector |
| $\tilde{\mathbf{A}}$ | New Adjacency matrix |
| $E'$ | New edges |
| $H_{uv}$ | Hitting time between $u$ and $v$ |
| $CT_{uv}$ | Commute time: $CT_{uv} = H_{uv} + H_{vu}$ |
| $R_{uv}$ | Effective resistance: $R_{uv} = CT_{uv}/vol(G)$ |
| $\mathbf{Z}$ | Matrix of commute times embeddings for all nodes in $G$ |
| $\mathbf{z}_u$ | Commute times embedding of node $u$ |
| $\mathbf{T}^{CT}$ | Resistance diffusion or commute times diffusion |
| $\mathbf{R}(\mathbf{Z})$ | Pairwise Euclidean distance of embedding $\mathbf{Z}$ divided by $vol(G)$ |
| $\mathbf{S}$ | Cluster assignment matrix: $\mathbf{S} \in \mathbb{R}^{n \times 2}$ |
| $\mathbf{T}^{GAP}$ | GAP diffusion |
| $\mathbf{e}_u$ | Unit vector with unit value at $u$ and 0 elsewhere |
| $\nabla_{\tilde{\mathbf{A}}}\lambda_2$ | Gradient of $\lambda_2$ wrt $\tilde{\mathbf{A}}$ |
| $[\nabla_{\tilde{\mathbf{A}}}\lambda_2]_{ij}$ | Gradient of $\lambda_2$ wrt $\tilde{\mathbf{A}}_{uv}$ |
| $p_u$ | Node curvature: $p_u := 1 - \frac{1}{2}\sum_{u \sim w} R_{uv}$ |
| $\kappa_{uv}$ | Edge curvature: $\kappa_{uv} := 2(p_u + p_v)/R_{uv}$ |
| $\|$ | Concatenation |

## A.1 Appendix A: CT-LAYER

### A.1.1 Notation

The Table 3 summarizes the notation used in the paper.

### A.1.2 Analysis of Commute Times rewiring

First, we provide an answer to the following question:

*Is resistance diffusion via $\mathbf{T}^{CT}$ a principled way of preserving the Cheeger constant?*

We answer the question above by linking Theorems 1 and 2 in the paper with the Lovász bound. The outline of our explanation follows three steps.

- **Proposition 1:** Theorem 1 (**Sparsification**) provides a principled way to bias the adjacency matrix so that the edges with the largest weights in the rewired graph correspond to the edges in graph's bottleneck.
- **Proposition 2:** Theorem 2 (**Cheeger vs Resistance**) can be used to demonstrate that increasing the effective resistance leads to a mild reduction of the Cheeger constant.
- **Proposition 3:** (Conclusion) The effectiveness of the above theorems to contain the Cheeger constant is constrained by the Lovász bound.

Next, we provide a thorough explanation of each of the propositions above.

**Proposition 1** (Biasing). *Let $G' = $ Sparsify$(G, q)$ be a sampling algorithm of graph $G = (V, E)$, where edges $e \in E$ are sampled with probability $q \propto R_e$ (proportional to the effective resistance). This choice is* necessary *to retain the global structure of $G$, i.e., to satisfy*

$$\forall \mathbf{x} \in \mathbb{R}^n : \ (1-\epsilon)\mathbf{x}^T \mathbf{L}_G \mathbf{x} \leq \mathbf{x}^T \mathbf{L}_{G'} \mathbf{x} \leq (1+\epsilon)\mathbf{x}^T \mathbf{L}_G \mathbf{x} , \tag{9}$$

*with probability at least $1/2$ by sampling $O(n \log n/\epsilon^2)$ edges , with $1/\sqrt{n} < \epsilon \leq 1$, instead of $O(m)$, where $m = |E|$. In addition, this choice* biases *the uniform distribution in favor of critical edges in the graph.*

*Proof.* We start by expressing the Laplacian $\mathbf{L}$ in terms of the edge-vertex incidence matrix $\mathbf{B}_{m \times e}$:

$$\mathbf{B}_{eu} = \begin{cases} 1 & \text{if } u \text{ is the head of } e \\ -1 & \text{if } u \text{ is the tail of } e \\ 0 & \text{otherwise .} \end{cases} \tag{10}$$

where edges in undirected graphs are counted once, i.e. $e = (u,v) = (v,u)$. Then, we have $\mathbf{L} = \mathbf{B}^T \mathbf{B} = \sum_e \mathbf{b}_e \mathbf{b}_e^T$, where $\mathbf{b}_e$ is a row vector (*incidence vector*) of $\mathbf{B}$, with $\mathbf{b}_{e=(u,v)} = (\mathbf{e}_u - \mathbf{e}_v)$. In addition, the Dirichlet energies can be expressed as norms:

$$\mathcal{E}(\mathbf{x}) = \mathbf{x}^T \mathbf{L} \mathbf{x} = \mathbf{x}^T \mathbf{B}^T \mathbf{B} \mathbf{x} = \|\mathbf{B}\mathbf{x}\|_2^2 = \sum_{e=(u,v) \in E} (\mathbf{x}_u - \mathbf{x}_v)^2 . \tag{11}$$

As a result, the effective resistance $R_e$ between the two nodes of an edge $e = (u,v)$ can be defined as

$$R_e = (\mathbf{e}_u - \mathbf{e}_v)^T \mathbf{L}^+ (\mathbf{e}_u - \mathbf{e}_v) = \mathbf{b}_e^T \mathbf{L}^+ \mathbf{b}_e \tag{12}$$

Next, we reformulate the spectral constraints in Eq. 9, i.e. $(1-\epsilon)\mathbf{L}_G \preccurlyeq \mathbf{L}_{G'} \preccurlyeq (1+\epsilon)\mathbf{L}_G$ as

$$\mathbf{L}_G \preccurlyeq \mathbf{L}_{G'} \preccurlyeq \Gamma \mathbf{L}_G , \Gamma = \frac{1+\epsilon}{1-\epsilon} . \tag{13}$$

This simplifies the analysis, since the above expression can be interpreted as follows: the Dirichlet energies of $\mathbf{L}_{G'}$ are lower-bounded by those of $\mathbf{L}_G$ and upper-bounded by $\Gamma$ times the energies of $\mathbf{L}_G$. Considering that the energies define hyper-ellipsoids, the hyper-ellipsoid associated with $\mathbf{L}_{G'}$ is between the hyper-ellipsoids of $\mathbf{L}_G$ and $\Gamma$ times the $\mathbf{L}_G$.

The hyper-ellipsoid analogy provides a framework to proof that the inclusion relationships are preserved under scaling: $M\mathbf{L}_G M \preccurlyeq M\mathbf{L}_{G'} M \preccurlyeq M\Gamma\mathbf{L}_G M$ where $M$ can be a matrix. In this case, if we set $M := (\mathbf{L}_G^+)^{1/2} = \mathbf{L}_G^{+/2}$ we have:

$$\mathbf{L}_G^{+/2} \mathbf{L}_G \mathbf{L}_G^{+/2} \preccurlyeq \mathbf{L}_G^{+/2} \mathbf{L}_{G'} \mathbf{L}_G^{+/2} \preccurlyeq \mathbf{L}_G^{+/2} \Gamma \mathbf{L}_G^{+/2} , \tag{14}$$

which leads to

$$\mathbf{I}_n \preccurlyeq \mathbf{L}_G^{+/2} \mathbf{L}_{G'} \mathbf{L}_G^{+/2} \preccurlyeq \Gamma \mathbf{I}_n . \tag{15}$$

We seek a Laplacian $\mathbf{L}_{G'}$ satisfying the *similarity constraints* in Eq. 13. Since $E' \subset E$, i.e. we want to remove structurally irrelevant edges, we can design $\mathbf{L}_{G'}$ in terms of considering *all* the edges $E$:

$$\mathbf{L}_{G'} := \mathbf{B}_G^T \mathbf{B}_G = \sum_e s_e \mathbf{b}_e \mathbf{b}_e^T \tag{16}$$

and let the similarity constraint define the sampling weights and the choice of $e$ (setting $s_e \geq 0$ properly). More precisely:

$$\mathbf{I}_n \preccurlyeq \mathbf{L}_G^{+/2} \sum_e \mathbf{b}_e \mathbf{b}_e^T \mathbf{L}_G^{+/2} \preccurlyeq \Gamma \mathbf{I}_n . \tag{17}$$

Then if we define $\mathbf{v}_e := \mathbf{L}_G^{+/2} \mathbf{b}_e$ as the *projected incidence vector*, we have

$$\mathbf{I}_n \preccurlyeq \sum_e s_e \mathbf{v}_e \mathbf{v}_e^T \preccurlyeq \Gamma \mathbf{I}_n . \tag{18}$$

Consequently, a spectral sparsifier must find $s_e \geq 0$ so that the above similarity constraint is satisfied. Since there are $m$ edges in $E$, $s_e$ must be zero for most of the edges. But, what are the best candidates to retain? Interestingly, the similarity constraint provides the answer. From Eq. 12 we have

$$\mathbf{v}_e^T \mathbf{v}_e = \|\mathbf{v}_e\|^2 = \|\mathbf{L}_G^{+/2} \mathbf{b}_e\|_2^2 = \mathbf{b}_e^T \mathbf{L}_G^+ \mathbf{b}_e = R_e . \tag{19}$$

This result explains why sampling the edges with probability $q \propto R_e$ leads to a ranking of $m$ edges of $G = (V, E)$ such that edges with large $R_e = \|\mathbf{v}_e\|^2$ are preferred[3].

Algorithm 1 implements a deterministic greedy version of $\texttt{Sparsify}(G, q)$, where we build incrementally $E' \subset E$ by creating a budget of decreasing resistances $R_{e_1} \geq R_{e_2} \geq \ldots \geq R_{e_{O(n \log n/\epsilon^2)}}$. $\quad\square$

Note that this rewiring strategy preserves the spectral similarities of the graphs, i.e. the global structure of $G = (V, E)$ is captured by $G' = (V, E')$.

Moreover, the maximum $R_e$ in each graph determines an upper bound on the Cheeger constant and hence an upper bound on the size of the graph's bottleneck, as per the following proposition.

---

**Algorithm 1:** GREEDYSparsify

---

**Input** : $G = (V, E), \epsilon \in (1/\sqrt{n}, 1], n = |V|$.
**Output** : $G' = (V, E')$ with $E' \subset E$ such that $|E'| = O(n \log n/\epsilon^2)$.

$L \leftarrow \text{List}(\{\mathbf{v}_e : e \in E\})$
$Q \leftarrow \text{Sort}(L, \text{descending}, \text{criterion}=\|\mathbf{v}_e\|^2)$ $\quad \triangleright$ Sort candidate edges by descending Resistance
$E' \leftarrow \emptyset$
$\mathcal{I} \leftarrow \mathbf{0}_{n \times n}$
**repeat**
$\quad$ $\mathbf{v}_e \leftarrow \text{pop}(Q)$ $\qquad\qquad\qquad\qquad\qquad\qquad$ $\triangleright$ Remove the head of the queue
$\quad$ $\mathcal{I} \leftarrow \mathcal{I} + \mathbf{v}_e \mathbf{v}_e^T$
$\quad$ **if** $\mathcal{I} \preccurlyeq \Gamma \mathbf{I}_n$ **then**
$\quad\quad$ $E' \leftarrow E' \cup \{e\}$ $\qquad\qquad\qquad\qquad$ $\triangleright$ Update the current budget of edges
$\quad$ **else**
$\quad\quad$ **return** $G' = (V, E')$
**until** $Q = \emptyset$

---

**Proposition 2** (Resistance Diameter). *Let G' = $\texttt{Sparsify}(G, q)$ be a sampling algorithm of graph $G = (V, E)$, where edges $e \in E$ are sampled with probability $q \propto R_e$ (proportional to the effective resistance). Consider the resistance diameter $\mathcal{R}_{diam} := \max_{u,v} R_{uv}$. Then, for the pair of $(u, v)$*

---

[3]Although some of the elements of this section are derived from [49], we note that the Nikhil Srivastava's lectures at The Simons Institute (2014) are by far more clarifying.

*does exist an edge $e = (u, v) \in E'$ in $G' = (V, E')$ such that $R_e = \mathcal{R}_{diam}$. A a result the Cheeger constant of $G$ $h_G$ is upper-bounded as follows:*

$$h_G \leq \frac{\alpha^\epsilon}{\sqrt{\mathcal{R}_{diam} \cdot \epsilon}} vol(S)^{\epsilon - 1/2}, \tag{20}$$

*with $0 < \epsilon < 1/2$ and $d_u \geq 1/\alpha$ for all $u \in V$.*

*Proof.* The fact that the maximum resistance $\mathcal{R}_{diam}$ is located in an edge is derived from two observations: a) Resistance is upper bounded by the shortest-path distance; and b) edges with maximal resistance are prioritized in (Proposition 1).

Theorem 2 states that any attempt to increase the graph's bottleneck in a multiplicative way (i.e. multiplying it by a constant $c \geq 0$) results in decreasing the effective resistances as follows:

$$R_{uv} \leq \left( \frac{1}{d_u^{2\epsilon}} + \frac{1}{d_v^{2\epsilon}} \right) \cdot \frac{1}{\epsilon \cdot c^2} \tag{21}$$

with $\epsilon \in [0, 1/2]$. This equation is called the *resistance bound*. Therefore, a multiplicative increase of the bottleneck leads to a quadratic decrease of the resistances.

Following Corollary 2 of [43], we obtain an upper bound of any $h_S$, i.e. the Cheeger constant for $S \subseteq V$ with $vol(S) \leq vol(G)/2$ – by defining $c$ properly. In particular we are seeking a value of $c$ that would lead to a contradiction, which is obtained by setting

$$c = \sqrt{\frac{\left( \frac{1}{d_{u^*}^{2\epsilon}} + \frac{1}{d_{v^*}^{2\epsilon}} \right)}{\mathcal{R}_{diam} \cdot \epsilon}}, \tag{22}$$

where $(u^*, v^*)$ is a pair of nodes with maximal resistance, i.e. $R_{u^*v^*} = \mathcal{R}_{diam}$.

Consider now any other pair of nodes $(s, t)$ with $R_{st} < \mathcal{R}_{diam}$. Following Theorem 2, if the bottleneck of $h_S$ is multiplied by $c$, we should have

$$R_{st} \leq \left( \frac{1}{d_s^{2\epsilon}} + \frac{1}{d_s^{2\epsilon}} \right) \cdot \frac{1}{\epsilon \cdot c^2} = \left( \frac{1}{d_s^{2\epsilon}} + \frac{1}{d_s^{2\epsilon}} \right) \cdot \frac{\mathcal{R}_{diam}}{\left( \frac{1}{d_{u^*}^{2\epsilon}} + \frac{1}{d_{v^*}^{2\epsilon}} \right)}. \tag{23}$$

However, since $\mathcal{R}_{diam} \leq \left( \frac{1}{d_{u^*}^{2\epsilon}} + \frac{1}{d_{v^*}^{2\epsilon}} \right)$ we have that $R_{st}$ can satisfy

$$R_{st} > \left( \frac{1}{d_s^{2\epsilon}} + \frac{1}{d_s^{2\epsilon}} \right) \cdot \frac{1}{\epsilon \cdot c^2} \tag{24}$$

which is a contradiction and enables

$$h_S \leq \frac{c}{vol(S)^{1/2 - \epsilon}} \iff |\partial S| \leq c \cdot vol(S)^{1/2 - \epsilon}. \tag{25}$$

Using $c$ as defined in Eq. 22 and $d_u \geq 1/\alpha$ we obtain

$$c = \sqrt{\frac{\left( \frac{1}{d_{u^*}^{2\epsilon}} + \frac{1}{d_{v^*}^{2\epsilon}} \right)}{\mathcal{R}_{diam} \cdot \epsilon}} \leq \sqrt{\frac{\alpha^\epsilon}{\mathcal{R}_{diam} \cdot \epsilon}} \leq \frac{\alpha^\epsilon}{\sqrt{\mathcal{R}_{diam} \cdot \epsilon}}. \tag{26}$$

Therefore,

$$h_S \leq \frac{c}{vol(S)^{1/2 - \epsilon}} \leq \frac{\frac{\alpha^\epsilon}{\sqrt{\mathcal{R}_{diam} \cdot \epsilon}}}{vol(S)^{1/2 - \epsilon}} = \frac{\alpha^\epsilon}{\sqrt{\mathcal{R}_{diam} \cdot \epsilon}} \cdot vol(S)^{\epsilon - 1/2}. \tag{27}$$

As a result, the Cheeger constant of $G = (V, E)$ is mildly reduced (by the square root of the maximal resistance). $\qquad \square$

**Proposition 3** (Conclusion). *Let $(u^*, v^*)$ be a pair of nodes (may be not unique) in $G = (V, E)$ with maximal resistance, i.e. $R_{u^*v^*} = \mathcal{R}_{diam}$. Then, the Cheeger constant $h_G$ relies on the ratio between the maximal resistance $\mathcal{R}_{diam}$ and its uninformative approximation $\left( \frac{1}{d_u^*} + \frac{1}{d_v^*} \right)$. The closer this ratio is to the unit, the easier it is to contain the Cheeger constant.*

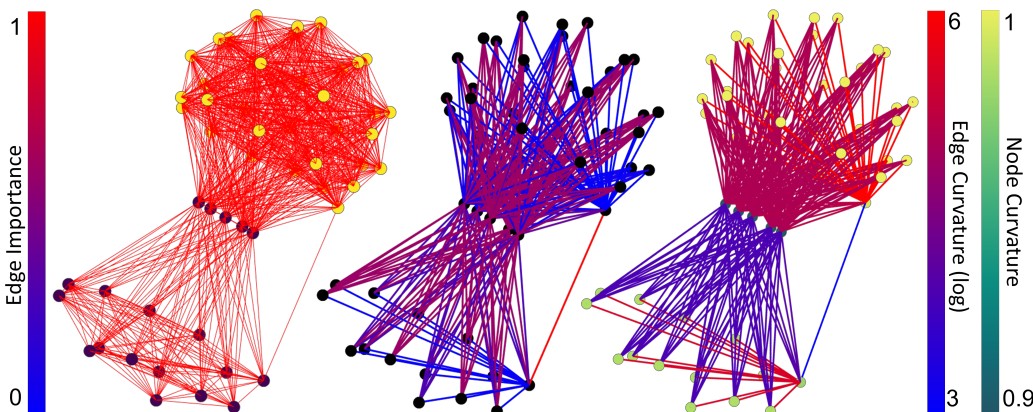

**Figure 5:** Left: Original graph with nodes colored as Louvain communities. Middle: $\mathbf{T}^{CT}$ learnt by CT-LAYER with edges colors as node importance [0,1]. Right: Node and edge curvature: $\mathbf{T}^{CT}$ using
$$p_u := 1 - \frac{1}{2} \sum_{u \sim w} \mathbf{T}^{CT}_{uv} \text{ and } \kappa_{uv} := 2(p_u + p_v)/\mathbf{T}^{CT}_{uv}$$
with edge an node curvatures as color. Graph from Reddit-B dataset.

*Proof.* The referred ratio above is the ratio leading to a proper $c$ in Proposition 2. This is consistent with a Lovász regime where the spectral gap $\lambda'_2$ has a moderate value. However, for regimes with very small spectral gaps, i.e. $\lambda'_2 \to 0$, according to the Lovász bound, $\mathcal{R}_{diam} \gg \left(\frac{1}{d^*_u} + \frac{1}{d^*_v}\right)$ and hence the Cheeger constant provided by Proposition 2 will tend to zero. $\square$

We conclude that we can always find an moderate upper bound for the Cheeger constant of $G = (V, E)$, provided that the regime of the Lovász bound is also moderate. Therefore, as the global properties of $G = (V, E)$ are captured by $G' = (V, E')$, a moderate Cheeger constant, when achievable, also controls the bottlenecks in $G' = (V, E')$.

Our methodology has focused on first exploring the properties of the commute times / effective resistances in $G = (V, E)$. Next, we have leveraged the spectral similarity to reason about the properties –particularly the Cheeger constant– of $G = (V, E')$. In sum, we conclude that resistance diffusion via $\mathbf{T}^{CT}$ is a principled way of preserving the Cheeger constant of $G = (V, E)$.

### A.1.3 Resistance-based Curvatures

We refer to recent work by Devriendt and Lambiotte [24] to complement the contributions of Topping et al. [20] regarding the use of curvature to rewire the edges in a graph.

**Theorem 3** (Devriendt and Lambiotte [24]). *The edge resistance curvature has the following properties: (1) It is bounded by $(4 - d_u - d_v) \leq \kappa_{uv} \leq 2/R_{uv}$, with equality in the lower bound iff all incident edges to $u$ and $v$ are cut links; (2) It is upper-bounded by the Ollivier-Ricci curvature $\kappa^{OR}_{uv} \geq \kappa_{uv}$, with equality if $(u, v)$ is a cut link; and (3) Forman-Ricci curvature is bounded as follows: $\kappa^{FR}_{uv}/R_{uv} \leq \kappa_{uv}$ with equality in the bound if the edge is a cut link.*

The new definition of curvature given in [20] is related to the resistance distance and thus it is learnable with the proposed framework (CT-LAYER). Actually, the Balanced-Forman curvature (Definition 1 in [20]) relies on the uniformative approximation of the resistance distance.

Figure 5 illustrates the relationship between effective resistances / commute times and curvature on an exemplary graph from the COLLAB dataset.

As seen in the Figure, effective resistances prioritize the edges connecting outer nodes with hubs or central nodes, while the intra-community connections are de-prioritized. This observation is consistent with the aforementioned theoretical explanations about preserving the bottleneck while breaking the intra-cluster structure. In addition, we also observe that the original edges between hubs have been deleted o have been extremely down-weighted.

Regarding curvature, hubs or central nodes have the lowest node curvature (this curvature increases with the number of nodes in a cluster/community). Edge curvatures, which rely on node curvatures, depend on the long-term neighborhoods of the connecting nodes. In general, edge curvatures can be seen as a smoothed version –since they integrate node curvatures– of the inverse of the resistance distances.

We observe that edges linking nodes of a given community with hubs tend to have similar edge-curvature values. However, edges linking nodes of different communities with hubs have different edge curvatures (Figure 5-right). This is due to the different number of nodes belonging to each community, and to their different average degree inside their respective communities (property 1 of Theorem 3).

Finally, note that the range of edge curvatures is larger than that of resistance distances. The sparsifier transforms a uniform distribution of the edge weights into a less entropic one: in the example of Figure 5 we observe a power-law distribution of edge resistances. As a result, $\kappa_{uv} := 2(p_u + p_v)/\mathbf{T}^{CT}_{uv}$ becomes very large on average (edges with infinite curvature are not shown in the plot) and a $\log$ scale is needed to appreciate the differences between edge resistances and edge curvatures.

## A.2 Appendix B: GAP-LAYER

### A.2.1 Spectral Gradients

The proposed GAP-LAYER relies on gradients wrt the Laplacian eigenvalues, and particularly the spectral gap ($\lambda_2$ for $\mathbf{L}$ and $\lambda'_2$ wrt $\mathcal{L}$). Although the GAP-LAYER inductively rewires the adjacency matrix $\mathbf{A}$ so that $\lambda_2$ is minimized, the gradients derived in this section may also be applied for gap maximization.

Note that while our cost function $L_{Fiedler} = \|\tilde{\mathbf{A}} - \mathbf{A}\|_F + \alpha(\lambda_2^*)^2$, with $\lambda_2^* \in \{\lambda_2, \lambda'_2\}$, relies on an eigenvalue, we *do not compute it explicitly*, as its computation has a complexity of $O(n^3)$ and would need to be computed in every learning iteration. Instead, we learn an approximation of $\lambda_2$'s eigenvector $\mathbf{f}_2$ and use its Dirchlet energy $\mathcal{E}(\mathbf{f}_2)$ to approximate the eigenvalue. In addition, since $\mathbf{g}_2 = \mathbf{D}^{1/2}\mathbf{f}_2$, we first approximate $\mathbf{g}_2$ and then approximate $\lambda'_2$ from $\mathcal{E}(\mathbf{g}_2)$.

**Gradients of the Ratio-cut Approximation.** Let $\mathbf{A}$ be the adjacency matrix of $G = (V, E)$; and $\tilde{\mathbf{A}}$, a matrix similar to the original adjacency but with minimal $\lambda_2$. Then, the gradient of $\lambda_2$ wrt each component of $\tilde{\mathbf{A}}$ is given by

$$\nabla_{\tilde{\mathbf{A}}}\lambda_2 := Tr\left[(\nabla_{\tilde{\mathbf{L}}}\lambda_2)^T \cdot \nabla_{\tilde{\mathbf{A}}}\tilde{\mathbf{L}}\right] = \text{diag}(\mathbf{f}_2\mathbf{f}_2^T)\mathbf{1}\mathbf{1}^T - \mathbf{f}_2\mathbf{f}_2^T ,\tag{28}$$

where $\mathbf{1}$ is the vector of $n$ ones; and $[\nabla_{\tilde{\mathbf{A}}}\lambda_2]_{ij}$ is the gradient of $\lambda_2$ wrt $\tilde{\mathbf{A}}_{uv}$. The above formula is an instance of the network derivative mining mining approach [46]. In this framework, $\lambda_2$ is seen as a function of $\tilde{\mathbf{A}}$ and $\nabla_{\tilde{\mathbf{A}}}\lambda_2$, the gradient of $\lambda_2$ wrt $\tilde{\mathbf{A}}$, comes from the chain rule of the matrix derivative $Tr\left[(\nabla_{\tilde{\mathbf{L}}}\lambda_2)^T \cdot \nabla_{\tilde{\mathbf{A}}}\tilde{\mathbf{L}}\right]$. More precisely,

$$\nabla_{\tilde{\mathbf{L}}}\lambda_2 := \frac{\partial \lambda_2}{\partial \tilde{\mathbf{L}}} = \mathbf{f}_2\mathbf{f}_2^T ,\tag{29}$$

is a matrix relying on an outer product (correlation). In the proposed GAP-LAYER, since $\mathbf{f}_2$ is approximated by:

$$\mathbf{f}_2(u) = \begin{cases} +1/\sqrt{n} & \text{if } u \text{ belongs to the first cluster} \\ -1/\sqrt{n} & \text{if } u \text{ belongs to the second cluster} \end{cases} ,\tag{30}$$

i.e. we discard the $O\left(\frac{\log n}{n}\right)$ from Eq. 30 (the non-liniarities conjectured in [17]) in order to simplify the analysis. After reordering the entries of $\mathbf{f}_2$ for the sake of clarity, $\mathbf{f}_2\mathbf{f}_2^T$ is the following block matrix:

$$\mathbf{f}_2\mathbf{f}_2^T = \left[\begin{array}{c|c} 1/n & -1/n \\ \hline -1/n & 1/n \end{array}\right] \text{ whose diagonal matrix is } \text{diag}(\mathbf{f}_2\mathbf{f}_2^T) = \left[\begin{array}{c|c} 1/n & 0 \\ \hline 0 & 1/n \end{array}\right]\tag{31}$$

Then, we have

$$\nabla_{\tilde{\mathbf{A}}}\lambda_2 = \left[\begin{array}{c|c} 1/n & 1/n \\ \hline 1/n & 1/n \end{array}\right] - \left[\begin{array}{c|c} 1/n & -1/n \\ \hline -1/n & 1/n \end{array}\right] = \left[\begin{array}{c|c} 0 & 2/n \\ \hline 2/n & 0 \end{array}\right]\tag{32}$$

which explains the results in Figure 1-left: edges linking nodes belonging to the same cluster remain unchanged whereas inter-cluster edges have a gradient of $2/n$. This provides a simple explanation for $\mathbf{T}^{GAP} = \tilde{\mathbf{A}}(\mathbf{S}) \odot \mathbf{A}$. The additional masking added by the adjacency matrix ensures that we do not create new links.

**Gradients Normalized-cut Approximation.** Similarly, using $\lambda'_2$ for graph rewiring leads to the following complex expression:

$$\nabla_{\tilde{\mathbf{A}}} \lambda'_2 := Tr \left[ (\nabla_{\tilde{\mathcal{L}}} \lambda_2)^T \cdot \nabla_{\tilde{\mathbf{A}}} \tilde{\mathcal{L}} \right] =$$

$$\mathbf{d}' \left\{ \mathbf{g}_2^T \tilde{\mathbf{A}}^T \tilde{\mathbf{D}}^{-1/2} \mathbf{g}_2 \right\} \mathbf{1}^T + \mathbf{d}' \left\{ \mathbf{g}_2^T \tilde{\mathbf{A}} \tilde{\mathbf{D}}^{-1/2} \mathbf{g}_2 \right\} \mathbf{1}^T \quad + \quad \tilde{\mathbf{D}}^{-1/2} \mathbf{g}_2 \mathbf{g}_2^T \tilde{\mathbf{D}}^{-1/2} . \quad (33)$$

However, since $\mathbf{g}_2 = \mathbf{D}^{1/2} \mathbf{f}_2$ and $\mathbf{f}_2 = \mathbf{D}^{-1/2} \mathbf{g}_2$, the gradient may be simplified as follows:

$$\nabla_{\tilde{\mathbf{A}}} \lambda'_2 := Tr \left[ (\nabla_{\tilde{\mathcal{L}}} \lambda_2)^T \cdot \nabla_{\tilde{\mathbf{A}}} \tilde{\mathcal{L}} \right] =$$

$$\mathbf{d}' \left\{ \mathbf{f}_2^T \tilde{\mathbf{D}}^{1/2} \tilde{\mathbf{A}}^T \mathbf{f}_2 \right\} \mathbf{1}^T + \mathbf{d}' \left\{ \mathbf{f}_2^T \tilde{\mathbf{D}}^{1/2} \tilde{\mathbf{A}} \mathbf{f}_2 \right\} \mathbf{1}^T \quad + \quad \tilde{\mathbf{D}}^{-1/2} \mathbf{f}_2 \mathbf{f}_2^T \tilde{\mathbf{D}}^{-1/2} . \quad (34)$$

In addition, considering symmetry for the undirected graph case, we obtain:

$$\nabla_{\tilde{\mathbf{A}}} \lambda'_2 := Tr \left[ (\nabla_{\tilde{\mathcal{L}}} \lambda_2)^T \cdot \nabla_{\tilde{\mathbf{A}}} \tilde{\mathcal{L}} \right] =$$

$$2 \mathbf{d}' \left\{ \mathbf{f}_2^T \tilde{\mathbf{D}}^{1/2} \tilde{\mathbf{A}} \mathbf{f}_2 \right\} \mathbf{1}^T + \tilde{\mathbf{D}}^{-1/2} \mathbf{f}_2 \mathbf{f}_2^T \tilde{\mathbf{D}}^{-1/2} . \quad (35)$$

where $\mathbf{d}'$ is a $n \times 1$ negative vector including derivatives of degree wrt adjacency and related terms. The obtained gradient is composed of two terms.

The first term contains the matrix $\tilde{\mathbf{D}}^{1/2} \tilde{\mathbf{A}}$ which is the adjacency matrix weighted by the square root of the degree; $\mathbf{f}_2^T \tilde{\mathbf{D}}^{1/2} \tilde{\mathbf{A}} \mathbf{f}_2$ is a quadratic form (similar to a Dirichlet energy for the Laplacian) which approximates an eigenvalue of $\tilde{\mathbf{D}}^{1/2} \tilde{\mathbf{A}}$. We plan to further analyze the properties of this term in future work.

The second term, $\tilde{\mathbf{D}}^{-1/2} \mathbf{f}_2 \mathbf{f}_2^T \tilde{\mathbf{D}}^{-1/2}$, downweights the correlation term for the Ratio-cut case $\mathbf{f}_2 \mathbf{f}_2^T$ by the degrees as in the normalized Laplacian. This results in a normalization of the Fiedler vector: $-1/n$ becomes $-\sqrt{d_u d_v}/n$ at the $uv$ entry and similarly for $1/n$, i.e. each entry contains the average degree assortativity.

### A.2.2 Beyond the Lovász Bound: the von Luxburg et al. bound

The Lovász bound was later refined by von Luxburg et al. [40] via a new, tighter bound which replaces $d_{min}$ by $d^2_{min}$ in Eq. 1. Given that $\lambda'_2 \in (0, 2]$, as the number of nodes in the graph ($n = |V|$) and the average degree increase, then $R_{uv} \approx 1/d_u + 1/d_v$. This is likely to happen in certain types of graphs, such as Gaussian similarity-graphs –graphs where two nodes are linked if the neg-exponential of the distances between the respective features of the nodes is large enough; $\epsilon$-graphs –graphs where the Euclidean distances between the features in the nodes are $\leq \epsilon$; and $k-$NN graphs with large $k$ wrt $n$. The authors report a linear collapse of $R_{uv}$ with the density of the graph in scale-free networks, such as social network graphs, whereas a faster collapse of $R_{uv}$ has been reported in community graphs –congruent graphs with Stochastic Block Models (SBMs) [44].

Given the importance of the effective resistance, $R_{uv}$, as a *global* measure of node similarity, the von Luxburg et al.'s refinement motivated the development of *robust effective resistances*, mostly in the form of $p-$resistances given by $R_{uv}^p = \arg \min_{\mathbf{f}} \{ \sum_{e \in E} r_e |f_e|^p \}$, where $\mathbf{f}$ is a unit-flow injected in $u$ and recovered in $v$; and $r_e = 1/w_e$ with $w_e$ being the edge's weight [50]. For $p = 1$, $R_{uv}^p$ corresponds to the shortest path; $p = 2$ results in the effective resistance; and $p \to \infty$ leads to the inverse of the unweighted $u$-$v$-mincut[4]. Note that the optimal $p$ value depends on the type of graph [50] and $p-$resistances may be studied from the perspective of $p-$Laplacians [45, 51].

While $R_{uv}$ could be unbounded by minimizing the spectral gap $\lambda'_2$, this approach has received little attention in the literature of mathematical characterization of graphs with small spectral gaps [52][53], i.e., instead of tackling the daunting problem of explicitly minimizing the gap, researchers in this field have preferred to find graphs with small spectral gaps.

---

[4]The link between CTs and mincuts is leveraged in the paper as an essential element of our approach.

### A.3 Appendix C: Experiments

In this section, we provide details about the graphs contained in each of the datasets used in our experiments, a detailed clarification about architectures and experiments, and, finally, report additional experimental results.

#### A.3.1 Datasets Statistics

Table 4 depicts the number of nodes, edges, average degree, assortativity, number of triangles, transitivity and clustering coefficients (mean and standard deviation) of all the graphs contained in each of the benchmark datasets used in our experiments. As seen in the Table, the datasets are very diverse in their characteristics. In addition, we use two synthetic datasets with 2 classes: Erdös-Rényi with $p_1 \in [0.3, 0.5]$ and $p_2 \in [0.4, 0.8]$ and Stochastic block model (SBM) with parameters $p_1 = 0.8$, $p_2 = 0.5$, $q_1 \in [0.1, 0.15]$ and $q_2 \in [0.01, 0.1]$.

**Table 4:** Dataset statistics. Parenthesis in *Assortativity* column denotes number of complete graphs (assortativity is undefined).

| | Nodes | Egdes | AVG Degree | Triangles | Transitivity | Clustering | Assortativity |
|---|---|---|---|---|---|---|---|
| REDDIT-B | 429.6 $\pm554$ | 497.7 $\pm622$ | 2.33 $\pm0.3$ | 24 $\pm41$ | 0.01 $\pm0.02$ | 0.04 $\pm0.06$ | -0.364 $\pm0.17$ (0) |
| IMDB-B | 19.7 $\pm10$ | 96.5 $\pm105$ | 8.88 $\pm5.0$ | 391 $\pm868$ | 0.77 $\pm0.15$ | 0.94 $\pm0.03$ | -0.135 $\pm0.16$ (139) |
| COLLAB | 74.5 $\pm62$ | 2457 $\pm6438$ | 37.36 $\pm44$ | $12\times10^4$ $\pm48\times10^4$ | 0.76 $\pm0.21$ | 0.89 $\pm0.08$ | -0.033 $\pm0.24$ (680) |
| MUTAG | 2.2 $\pm0.1$ | 19.8 $\pm5.6$ | 2.18 $\pm0.1$ | 0.00 $\pm0.0$ | 0.00 $\pm0.00$ | 0.00 $\pm0.00$ | -0.279 $\pm0.17$ (0) |
| PROTEINS | 39.1 $\pm45.8$ | 72.8 $\pm84.6$ | 3.73 $\pm0.4$ | 27.4 $\pm30$ | 0.48 $\pm0.20$ | 0.51 $\pm0.23$ | -0.065 $\pm0.2$ (13) |

In addition, Figure 6 depicts the histograms of the assortativity for all the graphs in each of the eight datasets used in our experiments. As shown in Table 4 assortativity is undefined in complete graphs (constant degree, all degrees are the same). Assortativity is defined as the normalized degree correlation. If the graph is complete, then both correlation and its variance is 0, so assortativity will be 0/0.

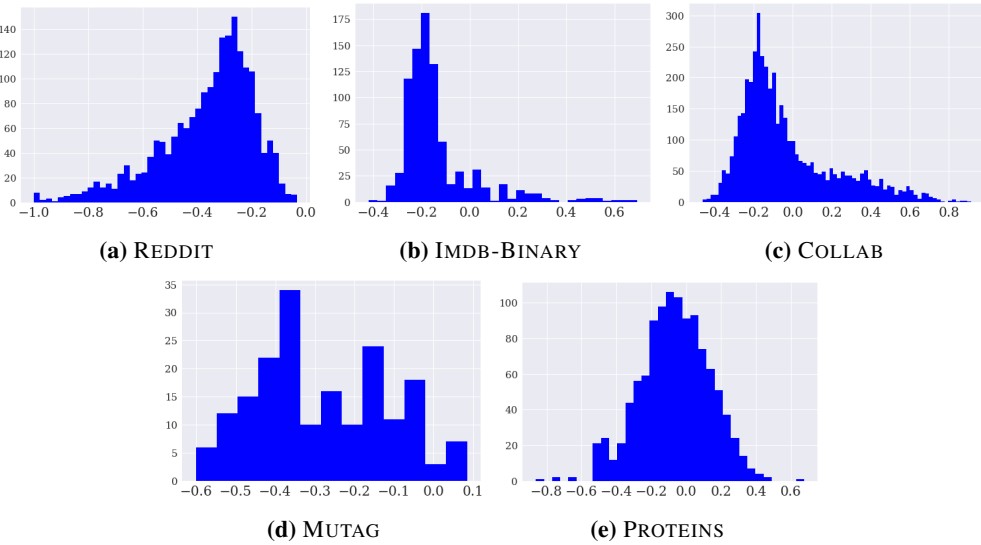

**(a)** REDDIT     **(b)** IMDB-BINARY     **(c)** COLLAB

**(d)** MUTAG     **(e)** PROTEINS

**Figure 6:** Histogram of the Assortativity of all the graphs in each of the datasets.

In addition, Figure 7 depicts the histograms of the average node degrees for all the graphs in each of the eight datasets used in our experiments. The datasets are also very diverse in terms of topology, corresponding to social networks, biochemical networks and meshes.

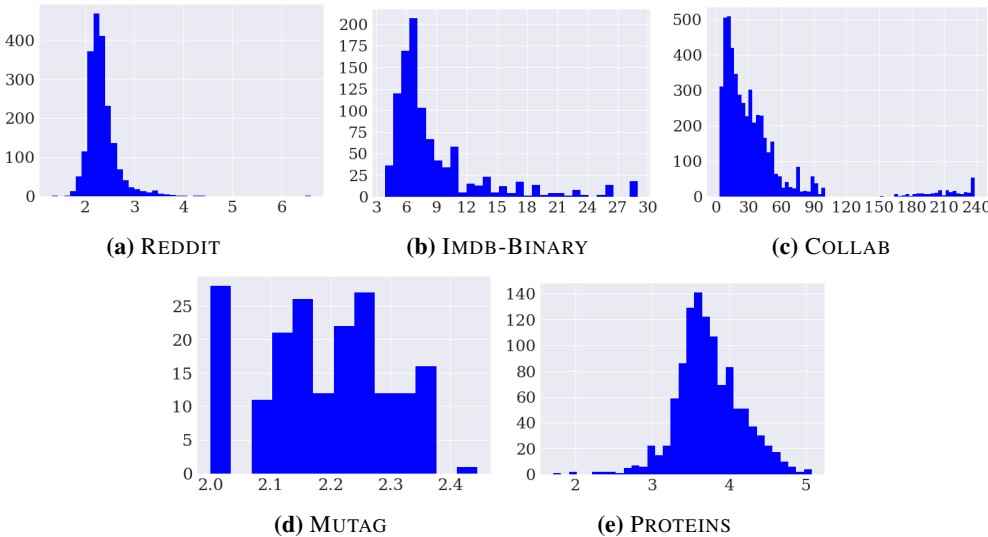

**Figure 7:** Degree histogram of the average degree of all the graphs in each of the datasets.

### A.3.2 Graph Classification GNN Architectures

Figure 8 shows the specific GNN architectures used in the experiments explained in section 4 in the manuscript. Although the specific calculation of $\mathbf{T}^{GAP}$ and $\mathbf{T}^{CT}$ are given in Theorems 2 and 1, we also provide a couple of pictures for a better intuition.

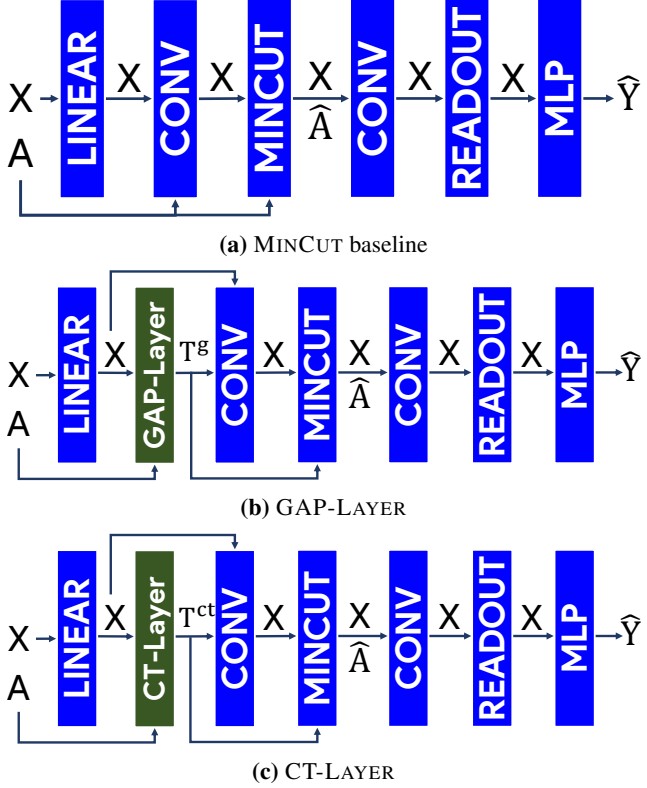

**Figure 8:** Diagrams of the GNNs used in the experiments.

### A.3.3 Training Parameters

The value of the hyperparameters used in the experiments are the ones by default in the code repository [5]. We report average accuracies and standard deviation on 10 random iterations, using different 85/15 train-test stratified split (we do not perform hyperparameter search), training during 60 epochs and reporting the results of the last epoch for each random run. We have used an Adam optimizer, with a learning rate of $5e-4$ and weight decay of $1e-4$. In addition, the batch size used for the experiments are shown in Table 5. Regarding the synthetic datasets, the parameters are: Erdös-Rényi with $p_1 \in [0.3, 0.5]$ and $p_2 \in [0.4, 0.8]$ and Stochastic block model (SBM) $p_1 = 0.8$, $p_2 = 0.5$, $q_1 \in [0.1, 0.15]$ and $q_2 \in [0.01, 0.1]$.

**Table 5:** Dataset Batch size

|  | Batch | Dataset size |
|---|---|---|
| REDDIT-BINARY | 64 | 1000 |
| IMDB-BINARY | 64 | 2000 |
| COLLAB | 64 | 5000 |
| MUTAG | 32 | 188 |
| PROTEINS | 64 | 1113 |
| SBM | 32 | 1000 |
| Erdös-Rényi | 32 | 1000 |

For the $k$-nn graph baseline, we choose $k$ such that the main degree of the original graph is maintained, i.e. $k$ equal to average degree. Our experiments also use 2 preprocessing methods DIGL and SDRF. Unlike our proposed methods, both SDRF [20] and DIGL [25] use a set of hyperparamerters to optimize for each specific graph, because both are also not inductive. This approach could be manageable for the task of node classification, where you only have one graph. However, when it comes to graph classification, the number of graphs are huge (5) and it is nor computationally feasible optimize parameters for each specific graph. For DIGL, we use a fixed $\alpha = 0.001$ and $\epsilon$ based on keeping the same average degree for each graph, i.e., we use a different dynamically chosen $\epsilon$ for each graph in each dataset which maintain the same number of edges as the original graph. In the case of SDRF, the parameters define how stochastic the edge addition is ($\tau$), the graph edit distance upper bound (number of iterations) and optional Ricci upper-bound above which an edge will be removed each iteration ($C^+$). We set the parameters $\tau = 20$ (the edge added is always near the edge of lower curvature), $C^+ = 0$ (to force one edge is removed every iteration), and number of iterations dynamic according to $0.7 * |V|$. Thus, we maintain the same number of edges in the new graph ($\tau = 20$ and $C^+ = 0$), i.e., same average degree, and we keep the graph distance to the original bounded by $0.7 * |V|$.

### A.3.4 Latent Space Analysis

In this section, we analyze the two latent spaces produced by the models.

- First, we compare the CT Embedding computed spectrally ($\mathbf{Z}$ in equation 2) with the CT Embedding predicted by our CT-LAYER ($\mathbf{Z}$ in definition 1) for a given graph, where each point is a node in the graph.

- Second, we compare the graph readout output for every model defined in the experiments (Figure 4) where each point is a graph in the dataset.

**Spectral CT Embedding vs CT Embeddings Learned by CT-LAYER .** The well-known embeddings based on the Laplacian positional encodings (PE) are typically computed beforehand and appended to the input vector $\mathbf{X}$ as additional features [35, 36]. This task requires an expensive computation $O(n^3)$ (see equation 2). Conversely, we propose a GNN Layer that learns how to predict the CT embeddings (CTEs) for unseen graphs (definition 1 and Figure 2) with a loss function that optimizes such CTEs. Note that we do not explicitly use the CTE features (PE) for the nodes, but we use the CTs as a new diffusion matrix for message passing (given by $\mathbf{T}^{\mathbf{CT}}$ in Definition 1). Note that we could also use $\mathbf{Z}$ as positional encodings in the node features, such that CT-LAYER may be seen as a novel approach to learn Positonal Encodings.

---

[5]https://github.com/AdrianArnaiz/DiffWire

In this section, we perform a comparative analysis between the spectral commute times embeddings (spectral CTEs, $\mathbf{Z}$ in equation 2) and the CTEs that are predicted by our CT-LAYER ($\mathbf{Z}$ in definition 1). As seen in Figure 9 (top), both embeddings respect the original topology of the graph, but they differ due to (1) orthogonality restrictions, and more interestingly to (2) the simplification of the original spectral loss function in Alev et al. [43]: the spectral CTEs minimize the trace of a quotient, which involves computing an inverse, whereas the CTEs learned in CT-LAYER minimize the quotient of two traces which is computationally simpler (see $L_{CT}$ loss in Definition 1). Two important properties of the first term in Definition 1 are: (1) the learned embedding $\mathbf{Z}$ has minimal Dirichlet energy (numerator) and (2) large degree nodes will be separated (denominator). Figure 9 (top) illustrates how the CTEs that are learned in CT-LAYER are able to better preserve the original topology of the graph (note how the nodes are more compactly embedded when compared to the spectral CTEs).

Figure 9 (bottom) depicts a histogram of the effective resistances or commute times (CTs) (see Section 3.2 in the paper) of the edges according to CT-LAYER or the spectral CTEs. The histogram is computed from the upper triangle of the $\mathbf{T^{CT}}$ matrix defined in Definition 1. Note that the larger the effective resistance of an edge, the more important that edge will be considered (and hence the lower the probability of being removed [54]). We observe how in the histogram of CTEs that are learned in CT-LAYER there is a 'small club' of edges with very large values and a large number of edges with low values yielding a power-law-like profile. However, the histogram of the effective resistances computed by the spectral CTEs exhibits a profile similar to a Gaussian distribution. From this result, we conclude that the use of $L_{CT}$ in the learning process of the CT-LAYER shifts the distribution of the effective resistances of the edges towards an asymmetric distribution where few edges have very large weights and a majority of edges have low weights.

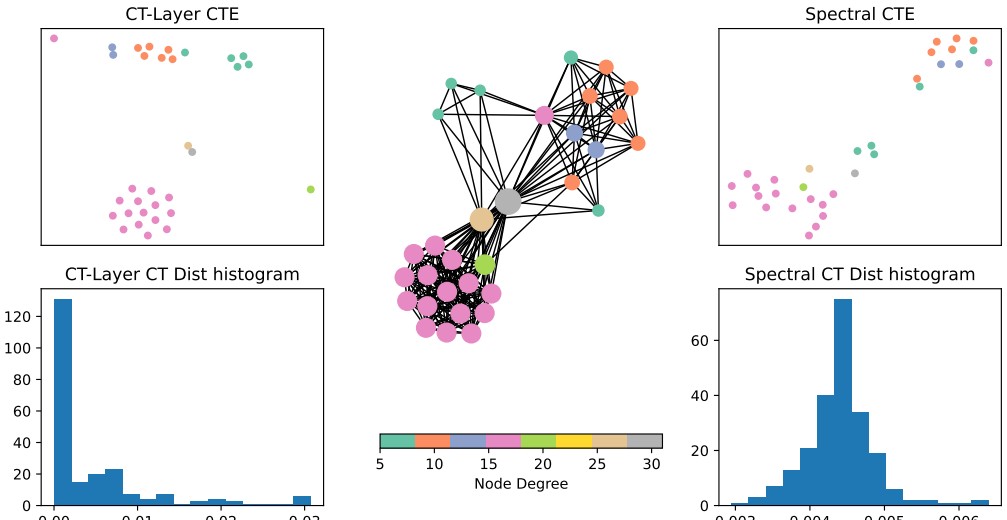

**Figure 9:** Top: CT embeddings predicted by CT-LAYER (left) and spectral CT embeddinggs (right). Bottom: Histogram of normalized effective resistances (i.e., CT distances or upper triangle in $\mathbf{T^{CT}}$) computed from the above CT embeddings. Middle: original graph from the COLLAB dataset. Colors correspond to node degree. CT-LAYER CTEs reduced from 75 to 32 dimensions using Johnson-Lindenstrauss. Finally, both CTEs reduced from 32 to 2 dimensions using T-SNE.

**Graph Readout Latent Space Analysis.** To delve into the analysis of the latent spaces produced by our layers and model, we also inspect the latent space produced by the models (Figure 4) that use MINCUTPOOL (Figure 8a), GAP-LAYER (Figure 8b) and CT-LAYER (Figure 8c). Each point is a graph in the dataset, corresponding to the graph embedding of the readout layer. We plot the output of the readout layer for each model, and then perform dimensionality reduction with TSNE.

Observing the latent space of the REDDIT-BINARY dataset (Figure 10), CT-LAYER creates a disperse yet structured latent space for the embeddings of the graphs. This topology in latent spaces show that this method is able to capture different topological details. The main reason is the expressiveness of the commute times as a distance metric when performing rewiring, which has been shown to be a

optimal metric to measure node structural similarity. In addition, GAP-LAYER creates a latent space where, although the 2 classes are also separable, the embeddings are more compressed, due to a more aggressive –yet still informative– change in topology. This change in topology is due to the change in bottleneck size that GAP-LAYER applies to the graph. Finally, MINCUT creates a more squeezed and compressed embedding, where both classes lie in the same spaces and most of the graphs have collapsed representations, due to the limited expressiveness of this architecture.

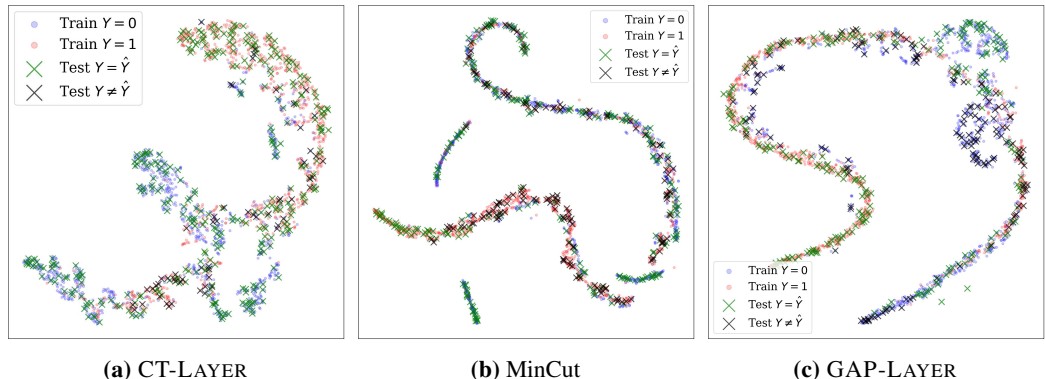

(a) CT-LAYER      (b) MinCut      (c) GAP-LAYER

**Figure 10:** REDDIT embeddings produced by GAP-LAYER (Ncut) CT-LAYER and MINCUT.

### A.3.5 Architectures and Details of Node Classification Experiments

The application of our framework for a node classification task entails several considerations. First, this first implementation of our method works with dense $\mathbf{A}$ and $\mathbf{X}$ matrices, whereas node classification typically uses sparse representations of the edges. Thus, the implementation of our proposed layers is not straightforward for sparse graph representations. We are planning to work on the sparse version of this method in future work.

Note that we have chosen benchmark datasets that are manageable with our dense implementation. In addition, we have chosen a basic baseline with 1 GCN layer to show the ability of the approaches to avoid under-reaching, over-smoothing and over-squashing.

The baseline GCN is a 1-layer-GCN, and the 2 compared models are:

- 1 CT-LAYER for calculating $\mathbf{Z}$ followed by 1 GCN Layer using $\mathbf{A}$ for message passing and $\mathbf{X} \parallel \mathbf{Z}$ as features. This approach is a combination of Velingker et al. [35] and our method. See Figure 11c.
- 1 CT-LAYER for calculating $\mathbf{T^{CT}}$ followed by 1 GCN Layer using that $\mathbf{T^{CT}}$ for message passing and $\mathbf{X}$ as features. See Figure 11b.

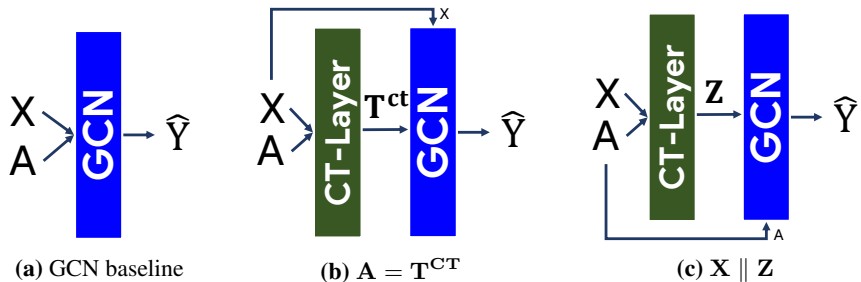

(a) GCN baseline      (b) $\mathbf{A} = \mathbf{T^{CT}}$      (c) $\mathbf{X} \parallel \mathbf{Z}$

**Figure 11:** Diagrams of the GNNs used in the experiments for node classification.

A promising direction of future work would be to explore how to combine both approaches to leverage the best of each of the methods on a wide range of graphs for node classification tasks. In addition, using this learnable CT distance for modulating message passing in more sophisticated ways is planned for future work.

### A.3.6 Analysis of Correlation between Structural Properties and CT-LAYER Performance

To analyze the performance of our model in graphs with different structural properties, we analyze the correlation between accuracy, the graph's assortativity, and the graph's bottleneck ($\lambda_2$) in COLLAB and REDDIT datasets. If the error is consistent along all levels of accuracy and gaps, the layer can generalize along different graph topologies.

As seen in Figure 14, Figure 12 (middle), and Figure 13 (middle), we do not identify any correlation or systematic pattern between graph classification accuracy, assortativity, and bottleneck with CT-LAYER-based rewiring, since the proportion of wrong and correct predictions are regular for all levels of assortativity and bottleneck size.

In addition, note that while there is a systematic error of the model over-predicting class 0 in the COLLAB dataset (see Figure 12), this behavior is not explained by assortativity or bottleneck size, but by the unbalanced number of graphs in each class.

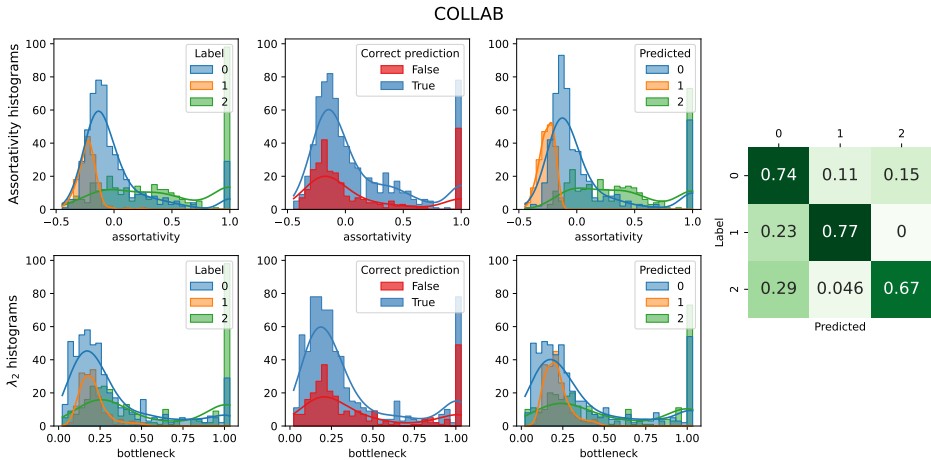

**Figure 12:** Analysis of assortativity, bottleneck and accuracy for COLLAB dataset. Top: Histograms of assortativity. Bottom: Histograms of bottleneck size ($\lambda_2$). Both are grouped by actual label of the graph (left), by correct or wrong predictions (middle) and by predicted label (right).

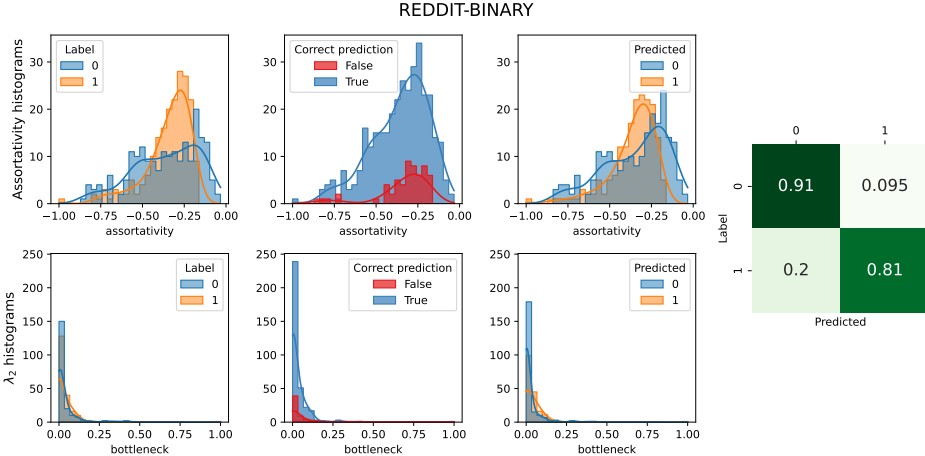

**Figure 13:** Analysis of assortativity, bottleneck and accuracy for REDDIT-B dataset. Top: Histograms of assortativity. Bottom: Histograms of bottleneck size ($\lambda_2$). Both are grouped by actual label of the graph (left), by correct or wrong predictions (middle) and by predicted label (right).

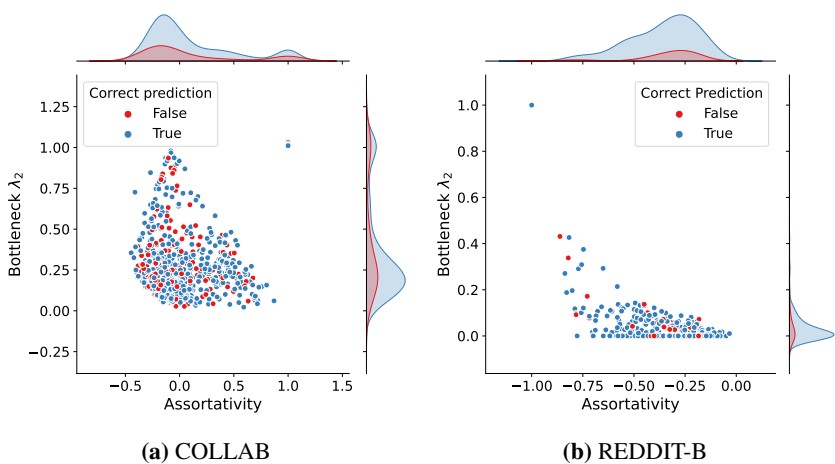

(a) COLLAB   (b) REDDIT-B

**Figure 14:** Correlation between assortativity, $\lambda_2$ and accuracy for CT-LAYER. Histograms shows that the proportion of correct and wrong predictions are regular for all levels of assortativity (x axis) and bottleneck size (y axis). For the sake of clarity, these visualizations, a and b, are the combination of the 2 histograms in the middle column of Figure 12 and Figure 13 respectively.

### A.3.7 Computing Infrastructure

Table 6 summarizes the computing infrastructure used in our experiments.

**Table 6:** Computing infrastructure.

| Component | Details |
| --- | --- |
| GPU | 2x A100-SXM4-40GB |
| RAM | 1 TiB |
| CPU | 255x AMD 7742 64-Core @ 2.25 GHz |
| OS | Ubuntu 20.04.4 LTS |

