# OpenReview forum: "DiffWire: Inductive Graph Rewiring via the Lovász Bound"
_logconference.io/LOG/2022/Conference — LoG 2022 Poster_

### Official Review · Reviewer_k8tG · 2022-10-20

**Overall Score:** 6
**Confidence:** 4

**Review:**

**Summary**

This paper proposes a novel aspect for graph rewiring task which is a well-estalished problem in the graph learning community. The proposed model leverages Lovász Bound as a fundamental tool, based on which some existing results are borrowed to shed lights on the task of graph rewiring. On top of the theory, two network layers on the basis of CT distance and spectral gap are proposed for improving graph rewiring. Finally, experiments on several real-world and synthetic datasets are conducted to verify the approach.

Overall, I vote for (weak) accepting. The proposed model is somewhat novel and theoretically grounded, and it aims to address an important and open research problem in the community. My major concern lies in the lack of empirical comparison with more graph rewiring baselines (see details below). Hopefully the authors can address my concern in the rebuttal period.

**Strengthes**

1. This paper studies a significant research problem in the ML community and provides a novel perspective for problem solving and analysis.

2. While the theory is already established in the literature, they are related and helpful for sheding insights on the problem and pave the way for the reasonable methods.

3. The proposed methods are reasonably validated by abaltion studies in the experiments.

4. The figures for illustrating the models are pretty nice and helpful for understanding the implementation of the models

**Weaknesses**

The weakness of this paper lies in the presentation of the model section. While the logic is clear, the technical contents seem fairly dense and not easy to follow. Also, while the related work section compares with graph structure learning, which aims to solve a similar research problem as is done by this work, more comparison with GSL methods in the experiment section is expected (see details below).

**Questions**

1. The proposed approach aims to improve the input structures via topological and spectral properties and consistency in a differentiable and inductive way, but the learnt structures are not optimized towards the downstream tasks. I wonder if the methods can truly benefit downstream prediction or it can merely maintain some good properties for the new structures. Also, the paper argues that the new model can learn inductive graph rewiring, but it is not clear what the `inductive' specifically means. More elaboration on this would be helpful if I miss something.

2. The experiments only compare the proposed models with several simplified variants as ablation studies. I would suggest adding more comparison with existing models on graph structure learning, which may help to make the contributions of this work more convincing. For example, comparison with simple similarity-based rewiring models (like KNN, kernel, etc.) or other learnable models (e.g., attention).

---

### Official Review · Reviewer_eQs8 · 2022-10-24

**Overall Score:** 8
**Confidence:** 3

**Review:**

########################################################################

Summary:

The paper proposes a novel graph rewiring scheme that is differentiable, maintains graph topology and doesn’t require hyper-parameter tuning. The author’s propose two different layers for MPNNs that satisfy these properties; A CT-layer which approximates commute times and re-weights edges based on this and a GAP-layer which rewires the adjacency matrix to minimise the spectral gap. The proposed layers perform well on the given benchmarks.

########################################################################

Reasons for Score:

Although the method proposed is novel, very interesting and performs well on the given benchmarks, I currently lean towards a reject as more needs to be done on describing how the method actually tackles the limitations of MPNNs discussed in the abstract and introduction  (over-smoothing, over-squashing and under-reaching). Additionally, the experimental analysis could be more convincing.

########################################################################

Strengths

- The theoretical justification for the layers in terms of differentiability and retaining the graph topology is very convincing. The use of the Lovász bound is novel and the framework given for rewiring is very exciting and I expect the community will find it useful and be able to build from it.
- The paper does a nice job of showing the relation between commute times, curvature and the spectral gap.
- The performance of these layers seems very promising.
- The authors do a good job of explaining the reasons for rewiring in general and its importance for the community.

########################################################################

Weaknesses:

- The authors state how other works have proposed edge sampling to avoid over-squashing (line 128) and removed and added edges to reduce over-squashing (line 134). However, the authors are not clear how their specific approach also mitigates these issues. Given that the abstract, introduction and related work focus a lot on over-squashing, under-reaching and over-smoothing, I think it would be extremely beneficial to outline how your specific layers can improve with regards to these issues either with a theoretical or experimental justification.
- The authors state the intention of the experimental section is to “shed light on the properties of both layers” - line 312. If this is the goal then I think more needs to be done in explaining the uncovered properties of the layers. Its not clear to me what properties you are trying to test (maybe write in the intro of the section why you specifically choose these datasets) and what insights you want to gain from this graph classification. The current results seem more like a limited analysis against a couple of previous rewiring approaches and the performance increase based on bottleneck properties is not rigorously analysed.

Some questions I might want to think about: Is improvement gained by rewiring or having the commute times as a feature (commute times improves GNNs here [1])? How are the layers changing the number of edges in practice across the datasets? are there no longer an exponential increase in nodes in the MPNN computational graphs to reduce over-squashing? Maybe you could put some of the latent space analysis in the main text? Is over-squashing/over-smoothing being reduced (you can increase number of layers without performance drop)?

########################################################################

Minor Suggestions/Thoughts

- Figure 1: This figure is quite hard to understand as there are a lot of edges in this graph. Maybe reducing this in the original graph, or including node degrees or making it clear the number of edges would be useful.
- “The larger the bottleneck, the more useful GAP-Layer is” - Line 364. Maybe you could convince me with a table of bottleneck vs accuracy. (Erdos-Renyi performs similarly between the two and the Lovász bound is very restrictive here).
- There is a lot of repeats in the introduction and related work - for example lines 94-115 seem to be very similar to the intro. These could be merged/removed I think.
- Line 329: You mention the runtime for graph classification for another rewiring approach. Maybe you could state the complexity of your layers and be clearer here how your approach is an improvement.
- Line 94: Maybe you could add references where MPNNs have shown competitive performances on these tasks.

########################################################################

Writing/Spelling

- Line 126 - Missing full-stop

[1] Affinity-Aware Graph Networks. Velingker et al

########################################################################
****Recommendation to Accept****
Having seen the authors response and the updates of the paper, I am changing my score to a clear accept.

---

### Official Review · Reviewer_Bnyp · 2022-10-25

**Overall Score:** 6
**Confidence:** 3

**Review:**

This paper proposes DiffWire, a fully differentiable, inductive, and parameter-free graph rewiring algorithm based on the Lovász bound. DiffWire uses the commute times as a relevance function for edge re-weighting, and proposes two types of new layers to either learn the commute times (CT-Layer) or adds a layer that optimizes the spectral gap for the network and the task (GAP-Layer). Experimental results show the proposed approach performs well on grass classification tasks.

Strengths:
- The paper is well-written with vigorous theoretical grounding. The connection between CT-Layer & GAP-Layer to the two sides of the Lovasz Bound and the graph’s spectral gap is interesting.
- Experimental results demonstrate the good performance of DiffWire on various graph classification datasets.
- CT-Layer and GAP-Layer’s performance differences on SBM verify the assumption of GAP-Layer being more suitable under the case that the Lovász bound is restrictive.

Weaknesses:
- Only two datasets have node features. As stated in the paper, DiffWire performs well on graphs with no node features as it can leverage the topology of the graphs. However, I am not sure if DiffWire can also utilize the node features well when they are informative, and asking for related justifications or additional experiments to be provided.

Questions:
- Why are experiments only conducted on graph classification tasks? Can the proposed framework also work for node classification tasks (like [1] mentioned in the paper)? It would be great if the authors can discuss this or provide experiments on node classification datasets.
- I am interested in if the graph homophily affects the effectiveness of the proposed method. A.3.1 claims to provide nodes, edges, average degree, assortativity, number of triangles, transitivity and clustering coefficients in table 3, but I cannot find the assortativity (assuming this refers to homophily). It would be desirable to include related data and discussion on graph homophily.
- I am unclear about how the statement "the smaller the graph’s bottleneck, the more useful the CT-Layer is" explains the better performance of CT-layer on COLLAB. Can this be further elaborated?

Overall this paper introduces a differentiable framework for graph rewiring with good theoretical and empirical support. Albeit having several confusions, I tend to accept this paper.

[1] Jake Topping, Francesco Di Giovanni, Benjamin Paul Chamberlain, Xiaowen Dong, and Michael M. Bronstein. Understanding over-squashing and bottlenecks on graphs via curvature. In *International Conference on Learning Representations*, 2022.

---

### Official Review · Reviewer_5GAb · 2022-10-27

**Overall Score:** 8
**Confidence:** 3

**Review:**

**Summary**:
This paper proposes two new GNN rewiring layer motivated by a bound from Lovász bounding the difference between the commute times between any two nodes and a term given through their degrees by the second eigenvalue of the normalised Laplacian and the minimum degree of the graph. Based on this bound, the authors propose to use either their first proposed layer (CT-layer) or the second proposed layer (GAP-layer). Instead of computing the eigenvectors involved in the CT-layer (with runtime roughly $O(kn^2)$ for $k$ eigenvectors) , they formulate a loss function that nudges the embedding vectors into this direction. They compare the two approaches with some graph rewiring baselines on typical benchmarks and achieve competitive performance.


**Main review**:

This paper provides two interesting approaches to the graph rewiring problem. However, my main critique is that the connection to the Lovász bound seems rather loose; one could derive and motivate the two proposed layers without it. Hence, focusing on it in the title is somewhat misleading. I am also not sure if I would call it "the Lovász bound", as it is not immediately clear which bound and property of Lovász is meant (e.g., it could also be Lovász' $\theta$ function). Either way, this work is probably interesting to the LoG community and tackles an important and recent problem, thus, I would for acceptance. However, I would recommend to motivate the two developed layers more clearly, as they seem rather ad-hoc, and also discuss how practitioners should choose between the two more clearly.


**Questions**:

The formulation of CT-layer is essentially the optimisation problem yielding the first $k$ eigenvectors of the Laplacian. Hence, the computed embeddings should be very similar to well-known embeddings based on (Laplacian) positional encodings. What is the difference to these kind of approaches?


**Further minor remarks**:

* line 38 typo: "to the capture" -> "to capture"
* line 43 and following: "k-hop" --> $k$-hop, "with k", "with $k$", etc.
* "Huang et al. [19] prove" --> they don't prove it (mathematically). Maybe better say "show"
* line 160 "is defined similarly" maybe "is defined analogously" is a more appropriate wording.
* Figure 1, colors in grey-scale unclear (the [0,1] vertical bar just looks black). I would recommend perceptually uniform colormaps (that work in grey-scale and colored).
* "semi-definite positive" --> "positive semi-definite"
* "spectral graph"--> probably meant "spectral gap"
* many mathematical functions / operators are formatted not properly. Please, use e.g., \DeclareMathOperator{..} for functions like MLP(), etc.

---

**rebuttal:**

I raised my score from weak accept to accept, as explained below.

---

### Meta-Review · Area_Chair_5522 · 2022-11-09

**Confidence:** 4
**Recommendation:** Accept

**Meta Review:**

This work proposes two new ways to perform graph rewiring to improve the performance of GNNs trained over graphs. The idea highly lies in spectral graph theory, which leverages commute-time minimization and eigengap maximization. The initial reviews lean toward borderline rejection mainly due to (a) the presentation issue of this work (dense math without a clear connection to the model itself); (b) limited experiments exclusively on small graph-level tasks where the over-smoothing/over-squashing issues are not that obvious. The authors' responses address a lot of such concerns so finally get all acceptance recommendations. I also recommend acceptance while I strongly suggest the authors can do the following changes in the final version, which were also claimed in their response.

1. A better explanation of the connection between the proposed method and the theory that drives the design. Most parts of Sec. 3 (Sec. 3.1, 3.2, cut approximation from Sec. 3.3) are well-established results in spectral graph theory. It is good to have them make the paper self-contained. However, it is just so much, which distracts readers. I suggest trimming that part or moving it to the appendix. In the main text, it is better to just focus on the architecture and why it helps with commute-time minimization and eigengap maximization. Theoretical results on commute time or eigengap can be mostly moved to the appendix, as they are not the contributions of this work and are kind of loosely connected with the design. For example, I see Eq. (4). If I understand it correctly, it is just to compute the eigenvalue decomposition of a normalized Laplacian matrix, which is not directly related to commuting time, because commute time has the inverse of eigenvalues as coefficients as shown in Eq.(3) but eigenvalue decomposition of a normalized Laplacian does not take such into account.

2. Merge the results on node-level tasks in the rebuttal into the manuscript. In the introduction, the authors argue that they may solve over-squashing, over-smoothing, non-reaching issues of GNNs. However, some of these issues, e.g., over-squashing, and non-reaching are actually crucial for node-level tasks over large graphs. They can be easily solved over small graphs, such as via transformer models where every two nodes linked or not can interact with each other. Some attention to learning graph structures [4] for node-level tasks may be also useful.

Also, I think the authors miss several earlier works on PEs, and structural features for graph topology and may want to discuss them in the final version. For example, in [1], the random-walk-based distance encoding has already covered the commute time between two nodes as a special case. In [2], how to use positional encoding in a stable and generalizable way is discussed. For this, I am actually curious about the stability of the proposed method in this work. In [3], a SOTA learnable PE is proposed. These PE works are particularly relevant in the sense that they construct structural features to improve expressive power. The rewiring idea of this paper shares a similar principle.

Also, although the authors state more expressive MPNN (see abstract), there are no arguments or analysis on the expressive of the proposed methods. Some results from [3] may be useful.

[1] Distance encoding: Design provably more powerful neural networks for graph representation learning. Li et al., NeurIPS 2020
[2] Equivariant and stable positional encoding for more powerful graph neural networks. Wang et al., ICLR 2022
[3] Sign and Basis Invariant Networks for Spectral Graph Representation Learning. Lim et al., arxiv.
[4] Learning Discrete Structures for Graph Neural Networks, ICML19

---

### Decision · Program_Chairs · 2022-11-22

**Decision:**

Accept (Poster)

**Comment:**

We discussed this among the PCs and agree with the AC assessment. We encourage authors to take the comments into account—in particular those pertaining to accessibility and framing in terms of related work.